# Efficacy and clinicogenomic correlates of response to immune checkpoint inhibitors alone or with chemotherapy in non-small cell lung cancer

Lingzhi Hong[1,2,12], Muhammad Aminu [2,12], Shenduo Li[3,12], Xuetao Lu[4], Milena Petranovic[5], Maliazurina B. Saad[2], Pingjun Chen [2], Kang Qin[1], Susan Varghese[1], Waree Rinsurongkawong[4], Vadeerat Rinsurongkawong[4], Amy Spelman[1], Yasir Y. Elamin [1], Marcelo V. Negrao [1], Ferdinandos Skoulidis[1], Carl M. Gay [1], Tina Cascone [1], Saumil J. Gandhi[6], Steven H. Lin [6], Percy P. Lee[6], Brett W. Carter[7], Carol C. Wu [7], Mara B. Antonoff [8], Boris Sepesi [8], Jeff Lewis[4], Don L. Gibbons[1], Ara A. Vaporciyan[8], Xiuning Le [1], J. Jack Lee [4], Sinchita Roy-Chowdhuri [9], Mark J. Routbort[9], Justin F. Gainor [10], John V. Heymach [1], Yanyan Lou[3,13], Jia Wu [1,2,13], Jianjun Zhang [1,11,13] ✉ & Natalie I. Vokes [1,11,13] ✉

The role of combination chemotherapy with immune checkpoint inhibitors (ICI) (ICI-chemo) over ICI monotherapy (ICI-mono) in non-small cell lung cancer (NSCLC) remains underexplored. In this retrospective study of 1133 NSCLC patients, treatment with ICI-mono vs ICI-chemo associate with higher rates of early progression, but similar long-term progression-free and overall survival. Sequential vs concurrent ICI and chemotherapy have similar long-term survival, suggesting no synergism from combination therapy. Integrative modeling identified PD-L1, disease burden (Stage IVb; liver metastases), and *STK11* and *JAK2* alterations as features associate with a higher likelihood of early progression on ICI-mono. *CDKN2A* alterations associate with worse long-term outcomes in ICI-chemo patients. These results are validated in independent external ($n = 89$) and internal ($n = 393$) cohorts. This real-world study suggests that ICI-chemo may protect against early progression but does not influence overall survival, and nominates features that identify those patients at risk for early progression who may maximally benefit from ICI-chemo.

While immune checkpoint inhibitors (ICIs) have changed the treatment of non-small cell lung cancer (NSCLC), only a minority of unselected patients experience long-term disease control[1]. Consequently, research efforts after the initial second- and first-line ICI-monotherapy approvals[2-6] have focused on identifying combination or biomarker-driven strategies to improve outcomes, leading to multiple approved first-line regimens incorporating combination chemotherapy[7-10] (ICI-chemo) or other immune checkpoint inhibitors[11,12]. However, while these trials increased the number of ICI-based first-line treatment options, they did not answer the central question as to who should receive combination therapy vs ICI monotherapy (ICI-mono), in large part because they compared ICI-combinations to chemotherapy

rather than head-to-head with ICIs alone. Underlying this confusion is an unclear mechanism of benefit; while some argue that ICI-chemo combinations synergistically increase the proportion of patients who benefit from ICIs[13,14], others contend that ICI and chemotherapy have an additive relationship, whereby ICI-chemo increases initial response rates simply due to patient populations with non-overlapping treatment sensitivities[15]. The implications of these distinct biological mechanisms are important; while a synergistic relationship would suggest that ICIs and chemotherapies should be given together, an additive relationship would suggest that, under appropriate circumstances, ICIs and chemotherapy can be given sequentially.

In the absence of randomized prospective clinical trials, retrospective analyses have focused on meta-analyses of published clinical trial data with conflicting results; some studies have shown improved outcomes with combination ICI-chemo, while others have shown no differential benefit[16–19]. The few published analyses of how these therapies perform in real-world patient cohorts have been limited by incomplete outcome annotations, small patient numbers, or focus on PD-L1 subgroups, with similarly conflicting results[20–22]. Further, a systematic analysis of the clinical and genomic features that predict preferential response to ICI-mono vs ICI-chemo has not yet been performed. Consequently, providers have little to guide them in identifying which patients require upfront combination therapy vs those who can be spared the excess toxicities; while many providers rely on PD-L1 ≥ 50% as a threshold for ICI-mono, PD-L1 is known to be an imperfect and potentially incomplete biomarker[23].

In this study, we utilize a large, clinically annotated cohort of ICI-treated NSCLC patients along with two validation cohorts to determine whether the addition of chemotherapy affects short- and long-term outcomes to ICI therapy in NSCLC, and to examine whether clinicogenomic features associate with ICI outcome to guide treatment selection.

## Results

### Clinical and molecular characteristics of study subjects

We identified 3584 patients aged 18 or older who were treated with ICIs between January 2014 and February 2020 in the MD Anderson Cancer Center (MDACC) GEMINI database. Of these, 1133 met the inclusion criteria (Methods; Supplementary Fig. 1) (MDACC-primary cohort). The clinical characteristics of study subjects are summarized in Table 1. Two validation cohorts—an external cohort of patients treated with first-line ICI regimens (Mayo cohort, n = 89), and a temporally distinct MDACC cohort of patients treated with first-line ICIs (MDACC-validation cohort, n = 393) were used to validate key findings (Supplementary Data 1). An additional, focused external validation cohort was obtained to assess overall survival in sequential vs concurrent therapy (MGH cohort, n = 193) (Methods). 675 patients were treated with first-line ICI, and 458 were treated in the second-line or later. PD-L1 proportion was high (≥50%), intermediate (1–49%), low (0 or <1%), and unknown in 240 (21.2%), 277 (24.4%), 254 (22.4%), and 362 (31.9%) patients, respectively. 735 patients in the MDACC-primary cohort had genomics data available; TP53 was the most frequently altered gene, altered in 60% of patients, followed by KRAS (37%), AR (21%), and STK11 (19%) (Supplementary Fig. 2; Supplementary Data 2). 273 patients in the MDACC-validation cohort and 89 in the Mayo cohort had available genomic data (Supplementary Data 3–4).

### Real world treatment patterns and selection of ICI-monotherapy vs ICI-chemotherapy

Across the entire MDACC-primary cohort (N = 1133), the median progression-free survival (PFS) was 5.6 months and median overall survival (OS) was 14.4 months, with a median follow-up of 26.0 months (range, 0–69.3), and 555 total deaths from any cause

**Table 1 | Clinicopathological characteristics of the entire cohort and first-line setting**

| Parameters | Total cohort, No. (%) | First line cohort, No. (%) |
|---|---|---|
| All | 1133 | 675 |
| **Age at ICI started** | | |
| Median, y, range | 64.9 (28.4–98.7) | 64.8 (31.5–91.5) |
| 18–65 | 569 (50.2) | 342 (50.7) |
| >65 | 564 (49.8) | 333 (49.3) |
| **Gender** | | |
| Male | 618 (54.5) | 374 (55.4) |
| Female | 515 (45.5) | 301 (44.6) |
| **Tobacco use** | | |
| Never | 227 (20) | 132 (19.6) |
| Former/current | 906 (80) | 543 (80.5) |
| **Histology** | | |
| LUAD | 862 (76.1) | 526 (77.9) |
| LUSC | 213 (18.8) | 122 (18.1) |
| Others | 58 (5.1) | 27 (4) |
| **Metastatic status at ICI start** | | |
| IVA | 486 (42.9) | 334 (49.5) |
| IVB | 647 (57.1) | 341 (50.5) |
| **Liver metastasis at ICI start** | | |
| No | 972 (85.5) | 585 (86.7) |
| Yes | 161 (14.2) | 90 (13.3) |
| **Brain metastasis at ICI start** | | |
| No | 824 (72.6) | 513 (76) |
| Yes | 309 (27.3) | 162 (24) |
| **ICI treatment** | | |
| ICI-mono | 726 (64.1) | 300 (44.4) |
| ICI-Chemo | 407 (35.9) | 375 (55.6) |
| **PD-L1 expression** | | |
| <1% | 254 (22.4) | 151 (22.4) |
| 1–49% | 277 (24.4) | 184 (27.3) |
| ≥50% | 240 (21.2) | 199 (29.5) |
| Unknown | 362 (31.9) | 141 (20.9) |

PD-L1 programmed death ligand 1, ICI immune checkpoint inhibitor, LUAD lung adenocarcinoma, LUSC lung squamous cell carcinoma, ICI-mono immune checkpoint inhibitor monotherapy, ICI-chemo immune checkpoint in combination with chemotherapy.

(49.0%). There was rapid uptake of first-line ICI-based treatment after 2016, and incorporation of chemotherapy-containing regimens after 2017 (Supplementary Fig. 3a–b). In the first-line cohort (n = 675), over half of patients (n = 375, 55.6%) were treated with ICI-chemo compared to ICI-mono (n = 300, 44.4%), with real-world outcomes that were comparable to or better than reported in first-line clinical trials (Supplementary Fig 3c–e). To understand which features associated with therapy selection, we stratified patients by initial treatment strategy (ICI-chemo vs ICI-mono) (Table 2). Compared to patients treated with first-line ICI-chemo, ICI-mono patients were more likely to be older (median age, 66.4 vs 63.9 years; P = 0.04), have squamous histology (28.3% vs 9.9%, P < 0.001), present without liver metastasis (11.3 vs 14.9%; P = 0.04), and have PD-L1 ≥ 50% (44.7% vs 17.3%, P < 0.001). We also observed a strong enrichment in STK11 alterations in ICI-chemo patients, and a weaker enrichment for TP53 alterations in ICI-mono patients (Supplementary Fig. 4a). Notably, STK11 alterations associated with lower PD-L1 expression, whereas TP53, TERT and NF1 alterations associated with higher PD-L1 expression, both on univariate and multivariate analysis (Supplementary Fig. 4b–d).

**Table 2 | Clinicopathologic features of patients treated with first-line ICI-mono vs ICI-chemo in MDACC (n = 675)**

| Parameters | No. (%) | | Odd ratio for ICI-chemo vs. ICI-mono | |
|---|---|---|---|---|
| | ICI-mono | ICI-chemo | Odd ratio (95% CI) | P (adjusted) |
| All | 300 (44.4) | 375 (55.6) | | |
| **Age at ICI started** | | | | |
| Median, y, range | 66.4 (33.5–91.5) | 63.9 (31.5–86.9) | | 0.04* |
| 18–65 | 139 (46.3) | 203 (54.1) | Reference | |
| >65 | 161 (53.7) | 172 (45.9) | 0.85 (0.60–1.21) | 0.374 |
| **Gender** | | | | |
| Male | 176 (58.7) | 198 (52.8) | Reference | |
| Female | 124 (41.3) | 177 (47.2) | 1.32 (0.91–1.90) | 0.138 |
| **Tobacco use** | | | | |
| Never | 51 (17) | 81 (21.6) | Reference | |
| Former/current | 249 (83) | 294 (78.4) | 0.83 (0.53–1.31) | 0.433 |
| **Histology** | | | | |
| LUAD | 201 (67) | 325 (86.7) | Reference | |
| LUSC | 85 (28.3) | 37 (9.9) | 0.23 (0.14–0.38) | 5.97e-9 |
| Others | 14 (4.7) | 13 (3.5) | 0.55 (0.22–1.34) | 0.192 |
| **Metastatic status at ICI start** | | | | |
| IVA | 163 (54.3) | 171 (45.6) | Reference | |
| IVB | 137 (45.7) | 204 (54.4) | 1.30 (1.86–1.97) | 0.209 |
| **Liver metastasis at ICI start** | | | | |
| No | 266 (88.7) | 319 (85.1) | Reference | |
| Yes | 34 (11.3) | 56 (14.9) | 1.84 (1.04–3.33) | 0.039 |
| **Brain metastasis at ICI start** | | | | |
| No | 242 (80.7) | 271 (72.3) | Reference | |
| Yes | 58 (19.3) | 104 (27.7) | 1.45 (0.92–2.31) | 0.113 |
| **PD-L1 expression** | | | | |
| <1% | 28 (9.3) | 123 (32.8) | reference | |
| 1–49% | 59 (19.7) | 125 (33.3) | 0.42 (0.24–0.72) | 0.002 |
| ≥50% | 134 (44.7) | 65 (17.3) | 0.08 (0.04–0.13) | 1.99e-16 |

Logistic regression models with adjusted effects were applied to calculate the odds ratio and *p* values.

*PD-L1* programmed death ligand 1, *ICI* immune checkpoint inhibitor, *LUAD* lung adenocarcinoma, *LUSC* lung squamous cell carcinoma.

*\*p*-value was calculated using Mann–Whitney *U* test.

## Outcomes to first-line ICI-mono vs ICI-chemo combination

To understand whether treatment choice affects outcome, we compared outcomes in first-line patients (*n* = 675) treated with ICI-mono vs ICI-chemo and observed no statistically significant difference in PFS or OS between treatment strategies (PFS: 7.2 vs 8.3, *P* = 0.5; OS: 24.7 vs 28.0, *P* = 0.2; Fig. 1a, b). To account for confounding by clinicopathologic variables underlying differential treatment selection, we also performed propensity score adjustment using the inverse probability of treatment weighting (IPTW) methodology (Methods; Supplementary Fig. 5), and observed no difference in first-line outcomes between ICI-mono vs ICI-chemo in the IPTW-adjusted cohort (Supplementary Fig. 6). These results held true when patients were

stratified by PD-L1 status (Supplementary Fig. 7), though there was a non-significant trend toward worse PFS in ICI-mono patients with PD-L1 < 50%. In both ICI-mono and ICI-chemo patients, high PD-L1 associated with improved PFS (Fig. 1c, d).

Response rates in clinical trials employing combination ICI-chemo have been higher than in ICI-mono trials[6,7], leading us to hypothesize that combination ICI-chemo might improve short-term rather than long-term outcomes. Consistent with this hypothesis, PFS rates at 3 months were higher with ICI-chemo than ICI-mono (3-month PFS, 85.2% vs 68.8%, *P* = 0.001). To more formally assess whether the effect of treatment changed over time, we modeled the cumulative hazard of ICI-chemo vs ICI-mono over time and observed a decreased hazard for progression associated with ICI-chemo in the first 3–6 months of therapy that subsequently disappeared (Fig. 1e). In fact, the ICI-chemo hazard increased relative to ICI-mono after 12 months, but this finding was not statistically significant. There was no significant difference in hazard for OS at any time point (Fig. 1f). When stratified by PD-L1 status, we observed similar trends in relative hazard in the PD-L1 ≥ 50% subgroup; there was no difference over time in the PD-L1 < 50% group (Supplementary. Fig 8).

We observed highly concordant results in the two validation cohorts, without significant difference in ICI-mono compared to ICI-chemo in either cohort (Supplementary Figs. 9–11). PFS rates at 3 months were also higher with ICI-chemo than ICI-mono in the MDACC-validation cohort (90% vs 76%, *P* < 0.001), but not in Mayo cohort (75% vs 77%, *P* = 0.826).

## Outcomes to sequential vs concurrent ICI and chemotherapy

One proffered rationale behind combining ICI and chemotherapy is a potential synergistic enhancement of efficacy[24,25]; however, a synergistic relationship has not been conclusively demonstrated. As the above outcome analyses suggested that ICI-chemo is protective against early progression but may not affect rates of long-term anti-tumor immunity, we next looked at whether sequencing of chemotherapy affects long-term OS, as synergy would be more likely from concurrent rather than sequential therapy. We compared OS in patients treated with ICI-mono with or without subsequent chemotherapy vs those treated concurrently (Fig. 2a), and observed no significant difference in OS between these three groups (*P* = 0.11) overall (Fig. 2b), nor when stratified into PD-L1 high or intermediate/low subgroups (Fig. 2c, d). Similar results were observed in our internal validation cohort (MDACC-validation; Supplementary Fig. 12a) as well as two external validation cohorts (Supplementary Fig. 12b, c). These findings suggest that patients who progress on initial ICI-mono but are treated with second-line chemotherapy have similar long-term outcomes to those treated with the upfront combination. In all three cohorts, patients treated with ICI-mono without subsequent chemotherapy trended toward worse OS in the first 12 months but no difference in long-term outcomes.

## Clinicopathologic predictors of benefit to ICI-mono vs ICI-chemo combination

We next sought to identify any clinical variables that associated with benefit to ICI and to determine whether these features were differentially predictive in the context of ICI-mono vs ICI-chemo. Focusing on the complete MDACC-primary cohort to increase power (*n* = 1133), on multivariate analysis in both the ICI-mono and ICI-chemo contexts, increased metastatic burden (stage IVb vs IVa and/or liver metastases) associated with worse progression-free survival, while PD-L1 ≥ 50% associated with improved PFS (ICI-mono: HR 0.45, 95% CI 0.28–0.72, *P* = 0.001; ICI-chemo: HR 0.36, 95% CI 0.18–0.72, *P* = 0.004) and OS (ICI-mono: HR 0.32, 95% CI 0.16–0.62, *P* = 0.001; ICI-chemo: HR 0.48, 95% CI 0.16–1.45, *P* = 0.19) (Fig. 3a, b); brain metastases and gender were not significant on any univariate analysis and were therefore excluded from the multivariable analysis (Supplementary Data 5). Smoking history (former/current smoker) trended toward improved

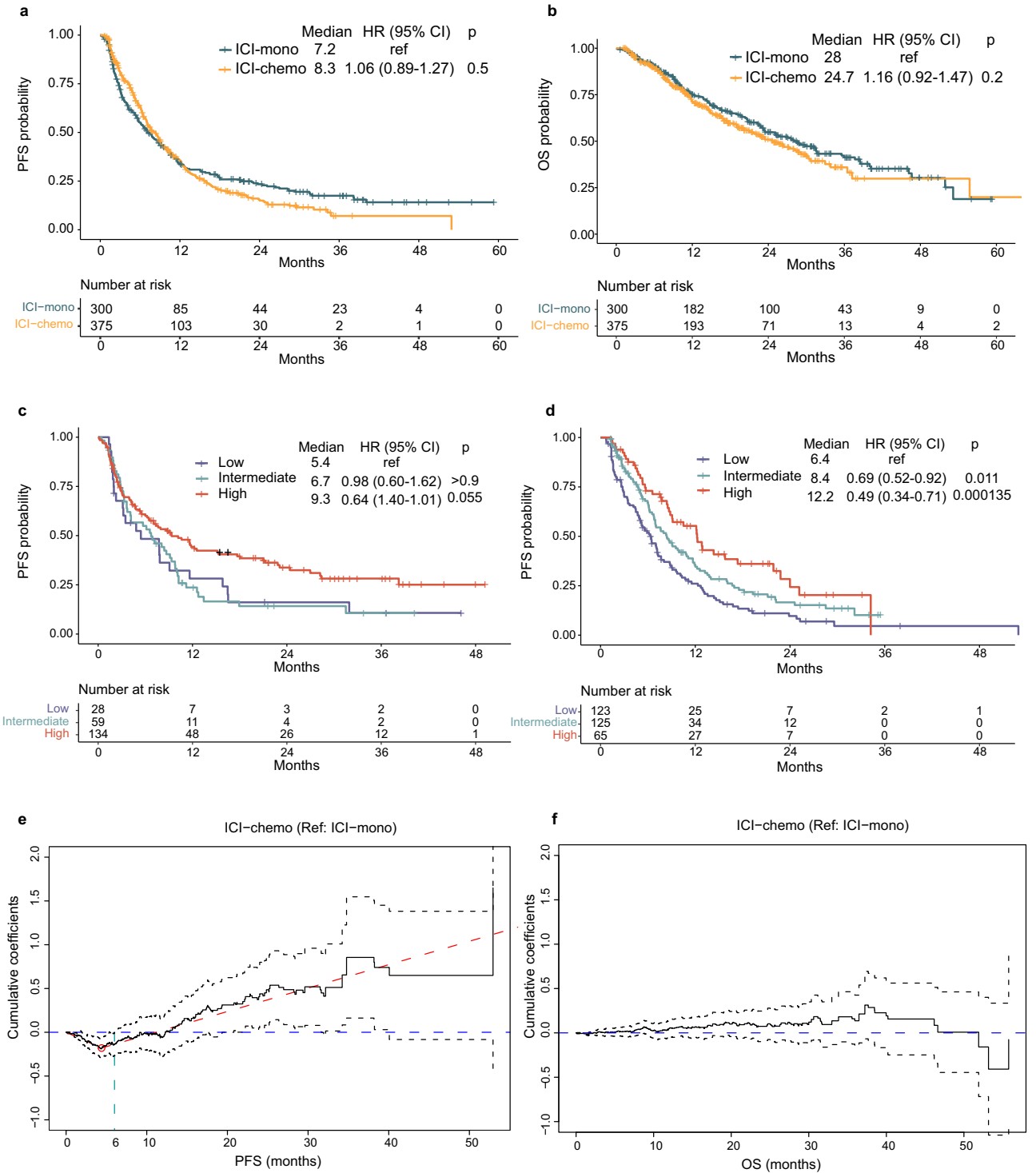

**Fig. 1 | Clinical outcomes in first-line patients treated with immune checkpoint inhibitors (ICI) as monotherapy (ICI-mono, *n* = 300) or with chemotherapy (ICI-chemo, *n* = 375) in the MDACC primary cohort.** Comparison of **a**, progression-free survival (PFS) and **b**, overall survival (OS) between ICI-mono vs ICI-chemo. **c**, PFS in ICI-mono stratified by PD-L1. **d** PFS in ICI-chemo stratified by PD-L1. Hazard ratio (HR) and *p* values were calculated using unadjusted cox proportional hazards regression models. **e** Aalen's additive hazard model on PFS and **f** on OS. Coefficient <0 favors ICI-chemo. Dashed gray lines indicate 95% confidence interval. Source data are provided as a Source Data file.

PFS (HR 0.79, 95% CI 0.61–1.04, *p*-value 0.089) in the ICI-mono group but was not significant in the ICI-chemo group or with OS. Treatment effect subgroup analysis to identify treatment-specific effects on PFS identified never smoking status as differentially favoring ICI-chemo (Fig. 3c). No variables associated with differential treatment effects on OS (Fig. 3d). Subgroup analysis confirmed that patients with never

smoking history treated with ICI-mono had the worst PFS, but no difference in OS compared to former/current smokers or those treated with ICI-chemo (Fig. 3e).

Allowing for decreased power, we observed very consistent trends in the first-line patient subset (*n* = 675) (Supplementary Fig. 13) and in the MDACC validation cohort (*n* = 393) (Supplementary Fig. 14), with

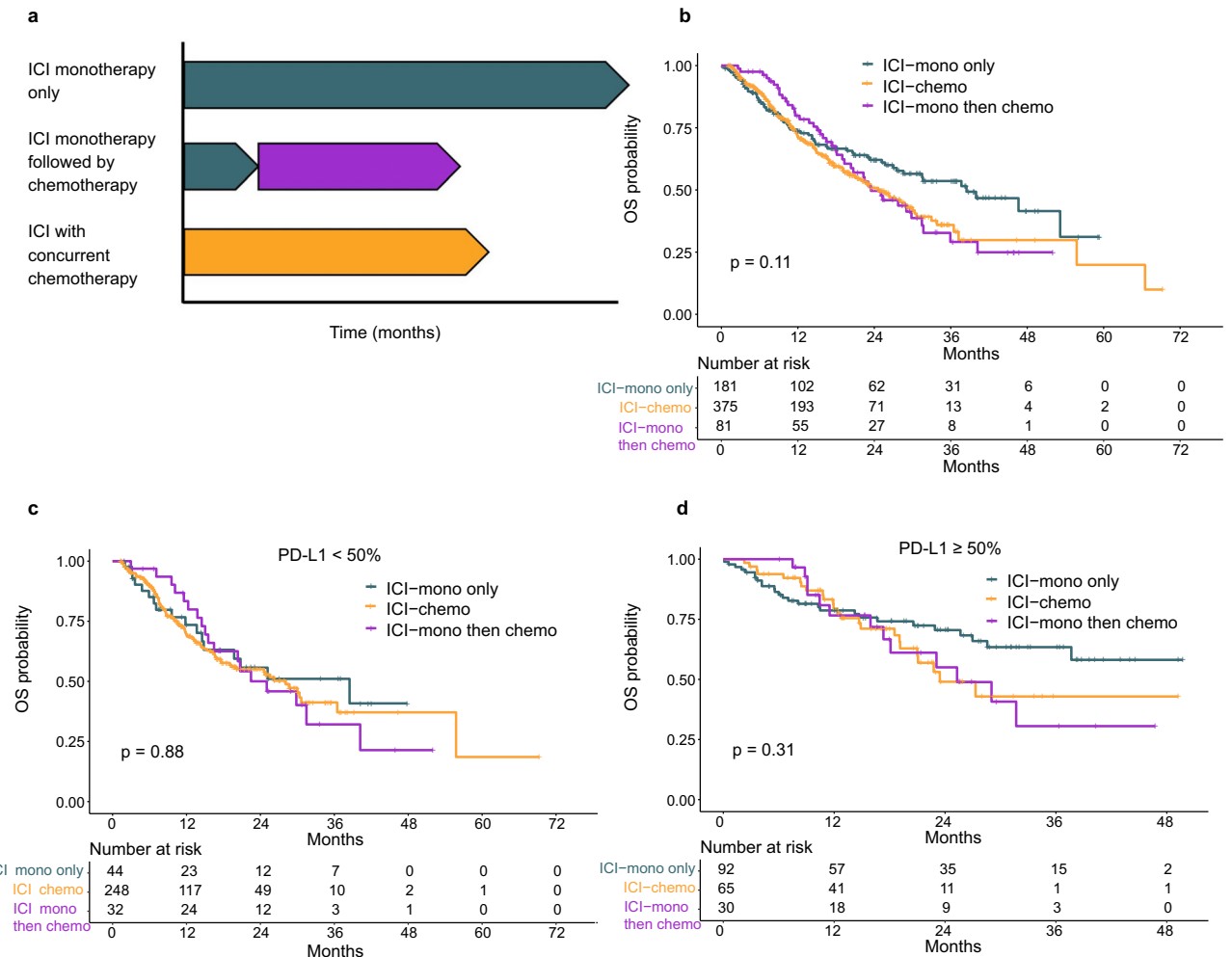

**Fig. 2 | Subgroup analyses of overall survival (OS) in MDACC-primary cohort first-line patients (*n* = 675) stratified by chemotherapy administration in the 2nd line. a** Analysis schema. **b** Kaplan–meier plot of OS of all first-line patients stratified by sequential vs combination chemotherapy; **c** OS in patients with PD-L1 intermediate (1%–49%) or low (0 or <1%); and **d** OS in patients with PD-L1 high (≥50%). *P* values were calculated using log-rank analysis. Source data are provided as a Source Data file. ICI-mono only: immune checkpoint inhibitor (ICI) monotherapy; ICI-chemo: concurrent ICI and chemotherapy; ICI-mono then chemo: sequential ICI monotherapy followed by chemotherapy.

the strongest associations arising from PD-L1 expression and markers of disease burden (stage IVb vs IVa; liver metastases). The Mayo validation cohort was not powered for subset analyses. Squamous histology associated with worse OS compared to adenocarcinoma in both the full and validation cohorts; while non-adenocarcinoma histology associated with worse PFS in the ICI-chemo context, the treatment effect was not significantly different (Fig. 3a–d; Supplementary Fig. 14), and treatment with ICI-mono vs ICI-chemo had no effect on histology-specific outcome (Fig. 3c; Supplementary Fig. 15).

Given that the most specific benefit to treatment with ICI-chemo vs ICI-mono appeared to be protection against progression in the first 3–6 months, we also examined which clinical features associated with progression at 3 months, and, similar to overall PFS, observed that low PD-L1 expression and higher disease burden (Stage IVb and liver metastases) associated with early progression (Supplementary Fig. 16a–b), with borderline significant treatment effects for smoking, liver metastases and stage IVb stage in favor of ICI-chemo (Supplementary Fig. 16c).

### Genomic predictors of early progression to ICI-mono and ICI-chemo

We next examined the genomic data to determine whether any genomic features associated with benefit to ICI-mono vs ICI-chemo, using the same analytic approach we applied to the clinical data. In patients treated with ICI-mono, alterations in *STK11*, *ERBB2*, *ARID1A,* and *CDK6* associated with a higher likelihood of progression at 3 months (Fig. 4a), whereas in ICI-chemo patients, only *STK11* associated with increased likelihood of 3-month progression (Fig. 4b). However, none of these genes had a significant treatment effect difference on subgroup analysis (Fig. 4c). Focusing on overall PFS, in ICI-mono treated patients, *CDK6*, *MET* and *ERBB2* all associated with shorter PFS and *KRAS* and *TP53* with longer PFS (Fig. 4d). In ICI-chemo treated patients, *CCND1* and *CDKN2A* alterations both associated with shorter PFS (Fig. 4e); interestingly, these genes also associated significantly with treatment strategy (Fig. 4f), with worse outcomes in ICI-chemo rather than ICI-mono treated patients with these alterations. On analysis of OS, no genes were significant in ICI-mono-treated patients, but *STK11* and *CCND1* associated with worse OS in ICI-chemo-treated patients (Supplementary Fig. 17).

To investigate the possible treatment-specific association with *CDKN2A* alterations, we compared PFS in patients stratified by treatment strategy and CDKN2A alteration status, focusing on deep deletions or mutations. We observed remarkably concordant results in both the primary cohort (Fig. 5a) and the Mayo validation cohort (Fig. 5b), where patients with *CDKN2A* alterations treated with ICI-chemo had worse outcomes relative to ICI-chemo/*CDKN2A* wild-type patients (MDACC: HR 0.452, 95% CI 0.229–0.892, *p* = 0.0221; Mayo: HR 0.646,

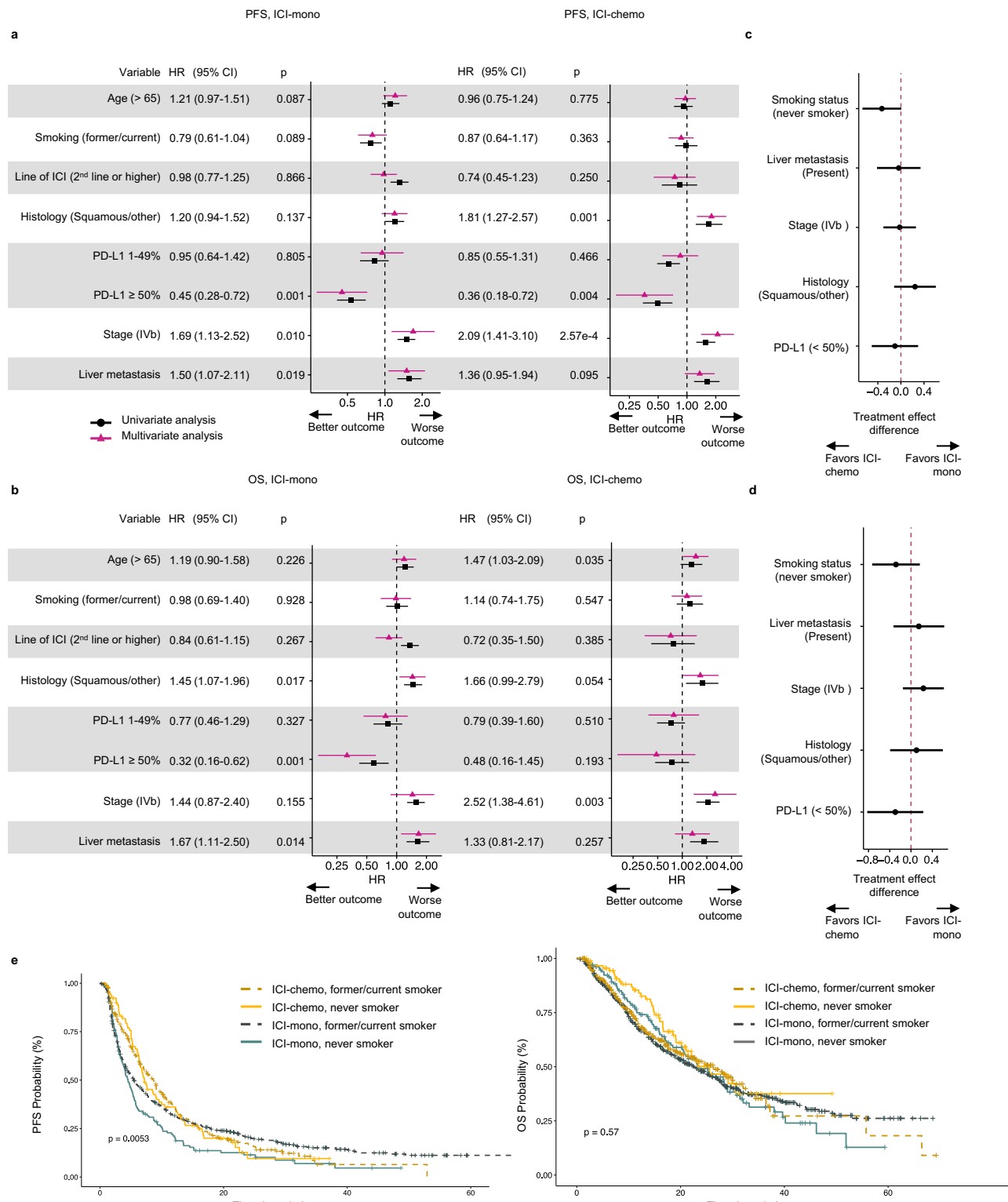

**Fig. 3 | Clinicopathological predictors of outcome by treatment strategy in the MDACC primary cohort (*n* = 1133).** Forest plot of clinicopathologic variables and association with **a**, progression-free survival (PFS) and **b**, overall survival (OS) on univariate (black) and multivariate (pink) analysis, stratified by ICI-monotherapy (ICI-mono, left panel) and ICI-chemotherapy (ICI-chemo, right panel). Data are presented as the hazard ratio with error bars showing 95% confidence interval. Cox proportional hazards regression models were applied to calculate the hazard ratio.

**c**, **d** Forest plot of difference in treatment effect between ICI-mono and ICI-chemo on **c** PFS and **d** OS. Data are presented as the treatment effect estimates with error bars showing the 95% confidence interval; subtee R package was used to generate treatment effect estimates. **e** Kaplan–meier plot comparing PFS (left panel) and OS (right panel) in patients treated with ICI-mono vs ICI-chemo, stratified by smoking status. Source data are provided as a Source Data file.

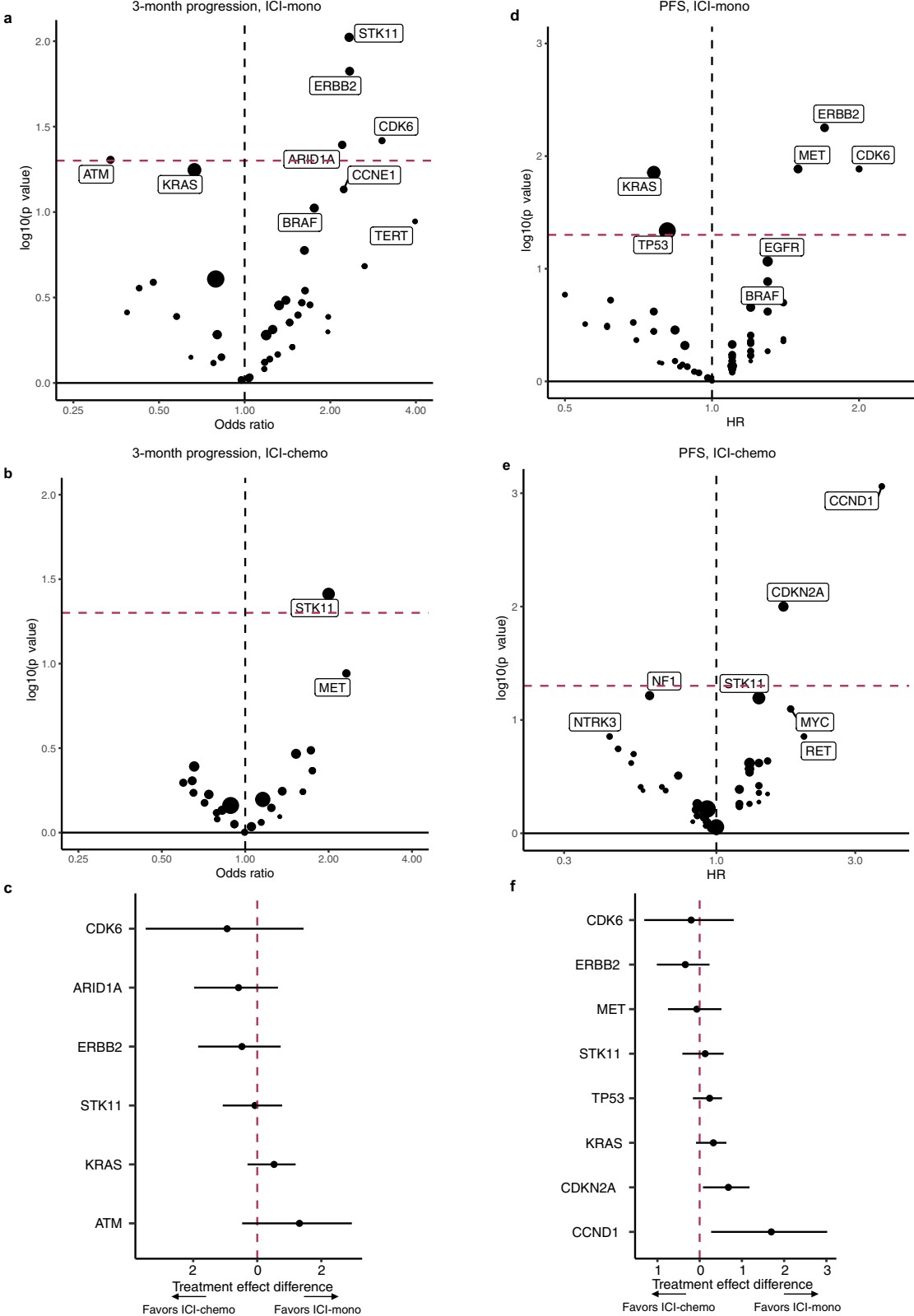

**Fig. 4 | Association between gene alterations and outcomes in the MDACC primary cohort (*n* = 735 with genomic data). a, b** Volcano plot from univariate logistic regression depicting odds ratio (x-axis) versus −log 10 (*P*-value) (y-axis) for 3-month progression in patients treated immune checkpoint inhibitors (ICI) as **a** monotherapy (ICI-mono) or **b** with chemotherapy (ICI-chemo); logistic regression models with unadjusted effect were applied to calculate the odds ratio and *p* values. **c** Treatment effect analyses of gene subgroups on 3-month progression. Data are presented as the treatment effect estimates with error bars showing the lower and

upper bounds of the 95% confidence interval. **d, e** Volcano plot from univariate cox regression depicting hazard ratio (x-axis) versus −log 10 (*P* value) (y-axis) for overall progression-free survival (PFS) in patients treated with **d** ICI-mono or **e** ICI-chemo; cox proportional hazards regression models with unadjusted effects were applied to calculate the hazard ratio and *p* values. **f** Treatment effect analyses of gene subgroups on PFS. Treatment effect estimates and the lower and upper bounds of the 95% confidence interval are shown as dots and whiskers. Source data are provided as a Source Data file.

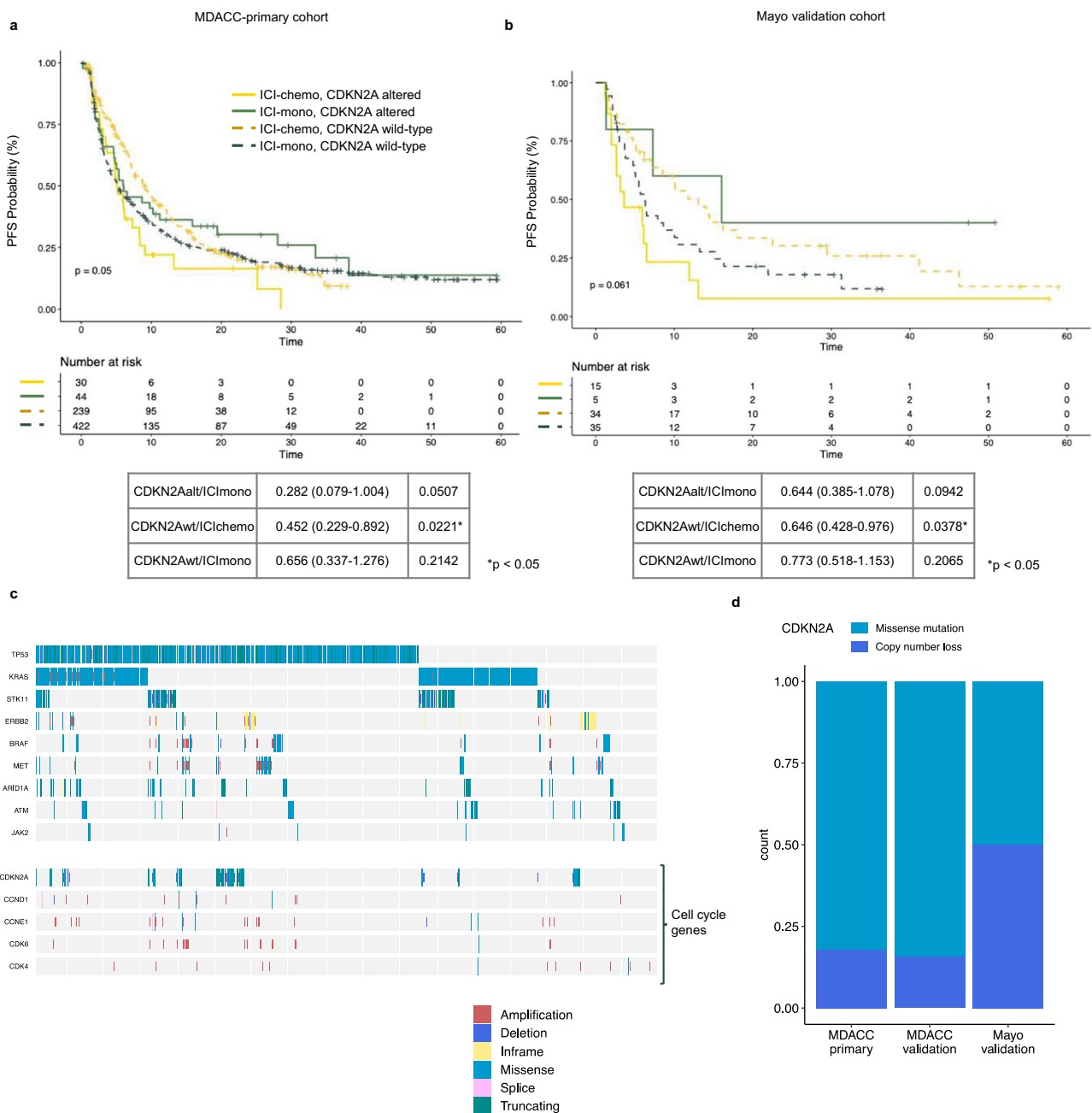

**Fig. 5 | Association between CDKN2A and progression-free survival (PFS).**
**a**, **b** Kaplan–Meier plot of PFS in patients treated with immune checkpoint inhibitor (ICI) monotherapy (ICI-mono) vs concurrent ICI-chemotherapy (ICI-chemo) stratified by CDKN2A in **a**, MDACC-primary ($n = 735$) and **b**, Mayo cohorts ($n = 89$); hazard ratio (HR) with 95% confident interval and p values within the tables were calculated using unadjusted cox proportional hazards regression models; $p$ values

in the survival plot were calculated using log-rank analysis. **c** Co-mutation plot of genes significantly associated with outcome, MDACC-primary cohort.
**d** Distribution of CDKN2A alterations (mutation and deletion) in MDACC-primary, MDACC-validation, and Mayo cohorts. Source data are provided as a Source Data file.

95% CI 0.428–0.976, $p = 0.0378$), and borderline significant worse PFS compared to patients with *CDKN2A* alterations treated with ICI-mono, consistent with a possible treatment specific effect. Similarly, in the MDACC validation cohort (Supplementary Fig. 18), *CDKN2A*-mutated patients treated with ICI-chemo had the worst PFS, though the PFS hazard ratio was only statistically significant when compared to *CDKN2A* wild-type/ICI-mono. As *CDKN2A* can be lost via mutation or copy number loss[26], with potentially different ICI-outcome associations, we catalogued the *CDKN2A* alterations detected in these cohorts, and observed a higher frequency of *CDKN2A* mutations than deletions detected, with cohort-specific differences likely driven by sequencing platform (Fig. 5c, d). Although analysis of outcome stratified by *CDKN2A*

loss event was limited by low numbers, in both the MDACC-primary and Mayo cohorts the results suggested that patients with *CDKN2A* mutations treated with ICI-mono had the best and most differential outcomes (Supplementary Fig. 19). Further focused analyses in additional datasets will be necessary to confirm these findings, along with the associations between outcome and *CCND1* and *CDK6*, which were not present in sufficient numbers in the validation cohorts for further analysis (MDACC-validation: *CDK6* altered $n = 6$, *CCND1* altered $n = 6$; Mayo: *CDK6* altered $n = 2$, *CCND1* altered $n = 5$).

*KRAS* alterations, which were one of the few genomic associations with improved PFS, associated with a history of tobacco exposure, adenocarcinoma histology, and trended toward higher PD-L1

expression in the MDACC-primary cohort, consistent with a more ICI-responsive clinical phenotype (Supplementary Table 1). However, *KRAS* mutations were not strongly associated with improved PFS in the MDACC validation cohort (HR 0.84, 95% CI 0.45–1.6, $P = 0.17$), reflecting the heterogeneity of this NSCLC subgroup[27–30]. STK11 co-mutation associated with worse outcomes in both cohorts, as previously reported (Supplementary Fig. 20)[31].

## Integrated clinicogenomic predictor of early progression to ICI-mono and ICI-chemo

Finally, to help guide provider treatment selection and account for the complicated inter-associations between genomic events and other clinicopathologic features such as PD-L1 (Supplementary Fig. 4), we integrated clinicopathologic variables with genomic features into a multivariate model. We used a feature selection approach to identify the most significant features driving early progression and to avoid over-fitting on the MDACC-primary cohort (training cohort) (Fig. 6a–f). The ranking score for each feature is listed in Supplementary Data 6 (ICI-mono cohort) and Supplementary Data 7 (ICI-chemo cohort). In the ICI-mono patients, liver metastases, metastatic stage (IVa vs IVb/c), PD-L1, *STK11*, *JAK2*, and histology were the top features associated with 3-month progression, and integration of these features into machine learning predictive models yielded a best AUC of 0.69–0.73 (Supplementary Data 6, Fig. 6a–c). Conversely, in ICI-chemo-treated patients, PD-L1 and *STK11* were the top features, but on the whole the predictive model performed less well (Fig. 6g), with a best AUC of 0.66, more heterogeneous model performance and no clear improvement with increasing feature number (Supplementary Data 7, Fig. 6d–f), suggesting that combination therapy renders these clinicogenomic features less predictive of 3-month PFS. The same models were then tested on the validation cohorts (Supplementary Figs. 21–22), with similarly improved performance in ICI-mono vs ICI-chemo and comparable overall model performance in all cohorts (ICI-mono, Mayo: AUC 0.88; MDACC-validation: AUC 0.73).

## Discussion

In this retrospective analysis of a large, real-world cohort of ICI-treated NSCLC patients, we determined that treatment with ICI-chemo compared to ICI-mono led to no difference in long-term progression-free or overall survival across PD-L1 levels. These findings were validated in independent internal and external cohorts. To date, there are few published data from similar real-world cohorts, and this analysis is one of few to include patients with squamous histology and PD-L1 < 50%. Several prior reports, including a recent analysis of real-world data focused on patients with PD-L1 ≥ 50%, demonstrated similar results, with comparable long-term outcomes in patients treated with ICI-chemo compared to ICI-mono[17,18,22]. In contrast, a recent abstract focused on patients with PD-L1 < 50% showed worse outcomes in IO-mono compared to IO-chemo[19]. We note that in our primary and internal validation cohorts there was separation in the PFS curves in patients with PD-L1 1–49%, but that this was not statistically significant, suggesting that there may be a weaker benefit to ICI-chemo in this subgroup that we were not powered to detect. Uniquely, we were also able to analyze whether treatment with sequential vs concurrent ICI and chemotherapy affected overall survival, and found no difference in long-term outcomes between either strategy.

However, while there were no long-term differences in outcome, our analysis of the hazard ratio over time suggests that there may be a benefit from the addition of chemotherapy in the prevention of early progression. Consistent with this interpretation, in our analysis of sequential vs concurrent therapy, there was an early drop-off in survival in patients treated with ICI-mono who did not receive subsequent chemotherapy, suggesting that at least some patients do not make it to second-line chemotherapy and may therefore have benefitted from the up-front combination. These findings have important translational

implications insofar as they suggest that the addition of chemotherapy to ICIs may be important in patients at high risk of early progression who are not salvageable with second-line treatment. A crucial clinical question, therefore, is understanding what features identify those patients at highest risk of early progression who might benefit from an initial ICI-chemo strategy. We attempt to address this question by building an integrative model for 3-month progression. This analysis capitalizes on our unique dataset, which includes both genomic data and in-depth clinical annotations, and identifies several high-risk features in addition to PD-L1, including higher disease burden, specifically liver metastases, *STK11* loss, and *JAK2* alterations as potential predictors of early progression. Importantly, these features were not as predictive of short-term progression in ICI-chemo, suggesting that combination therapy may mitigate the negative effect of these features on early progression. While our model will need to be validated in other datasets and prospectively, it provides early guidance as to which patients to preferentially select for combination ICI-chemo treatment.

Importantly, our results also suggest that chemotherapy does not synergistically increase the likelihood of long-term benefit. This interpretation is supported by the observation that many of the features that predict long-term PFS to ICI-mono, such as PD-L1, are the same as those that predict long-term PFS to ICI-chemo, indicating that patients who experience long-term benefits to ICIs do so because of their particular tumor-immune features rather than from the inclusion of chemotherapy. The observation that patients treated with ICIs and chemotherapy in sequence rather than simultaneously have comparable OS is further data in support of this interpretation. Consistent with our data, simulation analyses have shown that chemotherapy added to ICIs improves outcomes by combining therapies with non-overlapping populations of responders, consistent with an additive rather than synergistic benefit[32,33]. These findings have important therapeutic implications in that they suggest that patients without the high-risk features described above, or those who may be reasonably likely to achieve a second line of therapy, can be safely treated with upfront ICI-monotherapy and thereby avoid the toxicities of combination therapy. Whether the absence of benefit from chemotherapy over time arises in part from increased treatment-related toxicities cannot be determined from our data, but is an additional variable that will need to be explored as more real-world toxicity data becomes available.

Given these findings, disentangling any features that differentially affect sensitivity to ICI-mono vs ICI-chemo is challenging, as most of the long-term predictive clinicogenomic features reflect the likelihood of sustained ICI-mediated antitumor immunity rather than the likely minimal long-term effects of chemotherapy. Consistent with this, most of the predictive clinicogenomic features we identified did not differentially associate with treatment. However, we identified an association between *CDKN2A* alterations and differentially worse outcomes to ICI-chemotherapy, both in our primary and external validation cohorts. While *CDKN2A/B* loss has previously been associated with immunologically cold tumors and worse ICI outcomes[34,35], this association has not been consistent in NSCLC[36], nor is it known whether the immune phenotype is driven by *CDKN2A* loss or deletion of other genes in the 9p21 locus, which includes interferon alpha genes. We are not aware of prior data implicating *CDKN2A* alterations with differentially worse outcomes to ICI-chemotherapy, and these hypothesis-generating results will need further clinical and experimental validation. In addition, while the associations between ICI outcomes and amplifications in CDK6 and CCND1 will need to be validated in external cohorts, these preliminary findings suggest a possible role for dysregulation of cell cycle checkpoints in ICI-resistance, which to-date has been demonstrated in in vivo analyses[37,38] or smaller patient cohorts from other disease types[39], but not in NSCLC cohorts. Our data also confirms prior reports that never smokers may preferentially do worse when treated with ICI-mono[22,40], and validates previously reported associations between *STK11*[31,41], *ERBB2*[42] and worse ICI outcomes.

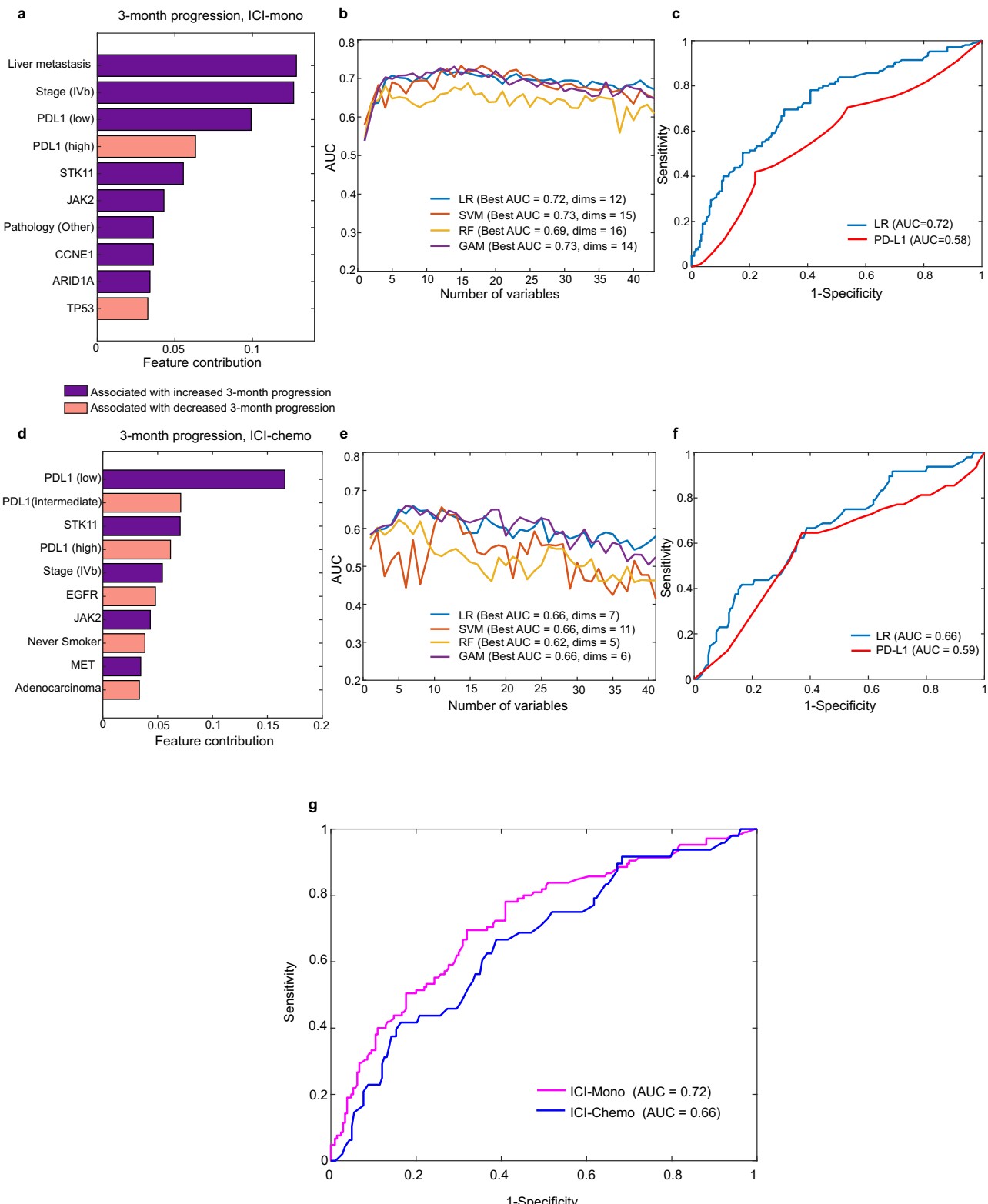

**Fig. 6 | Predictive models of 3-months progression on MDACC-primary cohort (training cohort) by logistic regression (LR), support vector machines (SVM), random forest (RF), and generalized additive model (GAM) for clinicopathological and genomic variables in immune checkpoint inhibitor (ICI) monotherapy (ICI-mono; a–c) and ICI with concurrent chemotherapy (ICI-chemo; d–f) treated patients. a** Bar chart showing the contribution of features with significant *p*-value from chi-squared feature selection; positive association with 3-month progression (worse effect) shown in purple, negative association (better effect) shown in pink. **b** Area under the curve (AUC) values generated by different model structures and number of included variables and **c** overall model performance relative to PD-L1 as benchmark in ICI-mono by receiver operator characteristic (ROC) curve. **d** Features ranked by significance in ICI-chemo cohort. **e** AUCs with increasing features and different model structures and **f** ROC for best performing LR model in ICI-chemo cohort. **g** ROC curves for the best performing LR models in ICI-mono vs ICI-chemo. Source data are provided as a Source Data file.

Our study is limited by the retrospective nature of the cohort, and outcomes may by confounded by the clinical features guiding therapeutic decisions. We also note that some subgroups, particularly the PD-L1 = 0% ICI-mono subgroup, are small and should be interpreted with caution. We attempted to address this limitation by using multivariate and propensity-score matched analyses, but acknowledge that there may be unaccounted for variables. We also note that certain genomic features, including TMB and copy number burden, are missing from this analysis. While smoking status may act as a clinical proxy for low vs high TMB subgroups and driver mutation status[23,40], understanding whether other genomic features associate with differential benefit to combination vs single-agent chemotherapy remains an important unanswered question. We consider the results presented herein to be exploratory and hypothesis-generating, and will require prospective validation prior to their clinical application.

In conclusion, this cohort of 1133 patients suggests that combination therapies may improve short-term but not long-term outcomes. These findings suggest that chemotherapy can be deployed in combination with ICI to improve early response rates in higher-risk patients, but also that, in appropriately selected patients, chemotherapy may be given sequentially with ICIs without compromising long-term outcomes. Patients with low PD-L1, liver metastases, stage IVb disease, or *STK11* alterations may particularly benefit from upfront combination treatment. Further investigation into *CDKN2A* alterations will help clarify the associated biology and determine whether patients with these alterations benefit from withholding combination chemotherapy. Additional studies will help validate these findings and refine clinical biomarkers to help guide optimized therapeutic selection to maximize benefit and minimize unnecessary toxicity.

## Methods

### Study population

#### MDACC Cohorts

**MDACC-Primary cohort.** We queried GEMINI, a University of Texas MD Anderson Cancer Center Lung Cancer Moon Shot funded internal database to identify patients treated with ICI who met the following criteria: (1) diagnosis of pathologically confirmed non-small cell lung cancer (NSCLC), including adenocarcinoma, squamous cell carcinoma, adenosquamous carcinoma, and NSCLC-not otherwise specified (NOS); (2) stage IV disease at the time of immune checkpoint inhibitor (ICI) start; and (3) received ≥2 cycles of ICI alone or with chemotherapy. Consistent with the FDA approvals, patients with known targetable *EGFR* or *ALK* alterations were excluded from the analysis. Clinical data were collected until September 10, 2020, when the dataset was locked for clinical outcome analysis.

**MDACC-Validation cohort.** Patients enrolled in GEMINI between March 2020–January 2022 (after the data lock for the MDACC-primary cohort) who were treated with ICI in the first-line setting who otherwise met the same inclusion criteria above were included in a temporally distinct validation cohort.

This study was approved by the institutional review board at MD Anderson Cancer Center.

**Mayo Cohort.** An external validation cohort of patients treated with ICIs at Mayo Clinic was identified. The same inclusion criteria were applied to the external validation cohort from Mayo Clinic. This cohort included patients who were seen at Mayo Clinic in Florida between 2017 and 2020. The updated data censor date is 7/1/2022. This study was approved by the institutional review board at Mayo Clinic and all patients provided written informed consent.

**MGH Cohort.** An additional validation cohort for the focused question of whether overall survival differed by treatment with concurrent vs sequential ICI and chemotherapy was obtained from Massachusetts General Hospital (MGH). This cohort consisted of patients with NSCLC treated with first-line ICIs at MGH between 01/2013-01/2020. As this cohort was assembled for radiographic analyses, patients without pre-treatment chest CTs or known ILD were (n = 8) were excluded. This study was approved by the institutional review board at MGH and all patients provided written informed consent.

### Clinical endpoints and annotations

Consistent with similar reports[22], real-world progression-free survival (PFS) and overall survival (OS) were defined as primary outcome measures[43,44]. Progression at 3 months was defined as a secondary outcome measure. PFS was defined as time from ICI initiation until progression or death. Disease progression was determined by the recorded assessment of the treating physician based on imaging reports of tumor growth or new disease sites, pathologic confirmation, or through clinical assessment of the treating physician. Patients who were alive without disease progression were censored at their last image assessment. OS was defined as time from ICI start until death from any cause. Patients alive at last follow-up were censored for the OS analysis. Metastatic sub-staging at time of ICI initiation was defined according to the 8th edition of the American Joint Committee on Cancer staging system.

### PD-L1 expression staining

PD-L1 expression was based on percentage of tumor cells expressing PD-L1 quantified by tumor proportion score (TPS), as assessed by Dako 22C3 internal stain at MD Anderson or reported by external lab reports. PD-L1 was described as high (TPS ≥ 50%), intermediate (1% ≤ TPS < 50%), low (TPS = 0 or <1%), and unknown.

### Genomic profiling

Somatic sequencing results from MD Anderson genomic profiling or external vendors from pathology reports and clinical notes were considered to exclude targetable EGFR or ALK alterations. For extended genomics analyses, samples with in-house somatic sequencing data were included. Mutational profiling was performed on formalin-fixed paraffin-embedded tumor tissue or blood samples as previously described[45,46]. The MD Anderson Molecular Diagnostics Laboratory tissue molecular profiling uses NGS-based analysis to detect mutations in 134 or 146 genes. Sequencing of circulating tumor DNA (ctDNA) was performed using the MD Anderson Liquid biopsy panel (70 genes) or the Guardant360 panel (74 genes) (Supplementary Data 8). Samples from any time point were considered to increase power, and a mutational event was considered present if it was detected in tissue and/or in blood. In patients with samples across multiple time points, the sample closest to the date of ICI start was selected. To increase the analysis consistency across sequencing panels, genomics analyses were limited to the 70 genes contained in the ctDNA panel (Supplementary Data 8). All reported non-synonymous mutational events were included; copy number alterations with log2 copy ratio <1 (deletion) or >5 (amplification) were included in the analysis. Due to smaller sequencing panel size[47], tumor mutational burden was not calculated.

### Statistical analysis

Cohort characteristics in patients treated with PD-(L)1 inhibitors alone (ICI-mono) and with PD-(L)1 inhibitors with chemotherapy (ICI-chemo) were compared using standard descriptive statistics. Categorical variables were reported as frequency and percentage and evaluated by Chi-Square test or Fisher's exact-test, as appropriate. Continuous variables were reported as medians and evaluated with the Mann–Whitney $U$ test.

The primary outcome analysis focused on comparing treatment outcomes in patients treated with first-line ICI-mono vs ICI-chemo, with planned subgroup analysis by PD-L1 grouping and smoking

status[22]. Survival estimates were obtained using the Kaplan–Meier method and crude differences between groups were assessed through the log-rank test. To assess whether the effect of treatment changed over time, Aalen's additive hazards model was applied[48]. To address possible systemic differences between the ICI-mono vs ICI-chemo group and reduce potential bias from these confounding factors, a propensity score-based analysis was performed. Propensity scores were estimated by regressing treatment assignment (ICI-mono vs ICI-chemo) on key prognostic variables (age, sex, tobacco use, histology, metastatic stage at ICI initiation, brain metastasis, liver metastasis, and PD-L1), and applied using the inverse probability of treatment weighting methodology (IPTW). Differences in primary outcomes between treatment groups were assessed in the IPTW-adjusted cohort. Post-weighting balance in covariates between treatment groups was evaluated using the standardized mean difference (SMD) approach, with an imbalance defined as SMD > 0.1.

In addition to the primary outcome analysis, focused exploratory analyses were performed to (1) identify any clinicogenomic features that associate with ICI outcome, and (2) determine whether any of these features were differentially predictive in the ICI-mono vs ICI-chemo context. Accordingly, Cox proportional hazards regression models were applied to identify clinicogenomic variables associated with PFS and OS in the ICI-mono and ICI-chemo contexts, and logistic regression model was applied to identify clinicogenomic variables associated with 3-month PFS. Clinical features were selected for multivariate analysis if they were significant on univariate analysis for PFS and OS, and pair-wise interaction terms that were significant were included in the multivariate model. Those variables that were significant ($p$-value < 0.05) on these regression analyses were selected for subgroup analysis to further interrogate the treatment effect. Exploratory subgroup and sensitivity analyses were not adjusted for multiple comparisons. The R package subtee[49] was used to generate treatment effect estimates. For association of PD-L1 with clinicogenomic variables, ordinal logistic regression using the MASS R package was performed, and significant variables on univariate analyses were assessed by multivariate analysis. Statistical analyses were performed using R software (Version 4.0.3), MATLAB (Version R2021a), and Python (Version 2.7.18), along with the R packages survminer (0.4.9), survival (3.2.13), MASS (7.3.54), dplyr (1.0.7), subtee (1.0.1), adjustedCurves (0.9.0). All hypotheses were two-sided, 95% confidence intervals are presented, and $p < 0.05$ was considered statistically significant.

### Clinicogenomic predictive model for progression at 3 months
Clinicopathologic features and genomic events that were present in ≥4 patients were included and annotated in both the MDACC cohort and Mayo cohort. The model was first generated in the MDACC-primary cohort, wherein a univariate feature ranking approach for classification using a Chi-square model was implemented to identify and rank relevant/nonredundant features associated with 3-month progression in the ICI-mono and ICI-chemo cohorts. This approach assigns a score (−log(p)) to each feature based on the p-value of the test statistic. Negative and positive associations were evaluated using mean square contingency coefficient (Phi Coefficient). After feature ranking, we implemented several classification models including logistics regression (LR), support vector machine (SVM), random forest (RF) and generalized additive model (GAM) to further examine the effectiveness of the selected (top ranked) features in predicting the 3-month progression. With the MDACC-primary cohort, we used k-fold ($k = 10$) cross validation technique to evaluate the performance of the different models. We applied the ICI-mono and ICI-chemo models to the MDACC-validation and Mayo cohorts to assess model performance in independent datasets.

### Reporting summary
Further information on research design is available in the Nature Portfolio Reporting Summary linked to this article.

## Data availability
Deidentified clinical data for patients in the MDACC-primary cohort reported in this study are available in Source data for Tables 1 and 2. Deidentified molecular data for patients in the MDACC-primary cohort are available in Source data for Supplementary Fig 2. The genomics data are included in this study as Supplementary Data 2 (MDACC-Primary cohort), Supplementary Data 3 (MDACC-Validation cohort) and Supplementary Data 4 (Mayo cohort). In addition, these data have been deposited in the Synapse.org database under accession code syn50877110 [DOI:10.7303/syn50877110]. The associated data are available under restricted access for noncommercial use. Access can be obtained by accepting the Synapse terms and conditions. Raw sequencing bam files from the clinical genomics panels are not available due to privacy laws; de-identified data for research use are available subject to institutional approval and can be requested from the corresponding author. In addition, anonymized data and the input for the predictive models are available at GitHub (https://github.com/nvokes/GEMINI_IO/tree/main, https://doi.org/10.5281/zenodo.7541973)[50]. The remaining data are available within the Article, Source Data, Supplementary Information, and Supplementary Data files. Source data are provided with this paper.

## Code availability
The code utilized for the predictive models in this study has been deposited in the repository available at GitHub (https://github.com/nvokes/GEMINI_IO/tree/main, https://doi.org/10.5281/zenodo.7541973)[50].

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

## Acknowledgements

N.I.V. is supported by a Conquer Cancer Young Investigator Award, the SITC Genentech Women in Cancer Immunotherapy Fellowship, and the Mark Foundation Damon-Runyon Physician Scientist Training award. This work was supported by the generous philanthropic contributions to The University of Texas MD Anderson Lung Moon Shot Program and the MD Anderson Cancer Center Support Grant P30 CA016672.

## Author contributions

L.H., J.Z., and N.I.V designed the study. L.H., S.L., M.P., K.Q., S.V., and W.R. collected data. L.H., M.A., X.L., M.B.S., P.C., J.J.L., Y.L., J.W., J.Z., and N.I.V. designed methodology and performed data interpretation. L.H., J.Z., and N.I.V. prepared the manuscript. All authors reviewed, revised, discussed results and contributed to the finalization of the manuscript.

## Competing interests

M.V.N. reports Research funding to institution from Mirati, Novartis, Alaunos, Checkmate, AstraZeneca, Pfizer, Genentech; and Consultant/ Advisory Board from Mirati, Novartis, Genentech, and Merck/MSD, outside the submitted work. Y.Y.E. reports research support from Spectrum, AstraZeneca, Takeda, Eli Lilly, Xcovery, and Tuning Point Therapeutics; and advisory role for AstraZeneca, Eli Lilly, Sanofi, BMS, Spectrum and Turning Point; and accommodation expenses from Eli Lilly. F.S. reports consulting fees and advisory roles from Amgen Inc., AstraZeneca Pharmaceuticals, Novartis, BeiGene, Tango Therapeutics, Calithera Biosciences, Navire Pharma, Medscape LLC, Intellisphere LLC, Guardant Health, and BergenBio; speaker fees from BMS, RV MaisPromocao Eventos LTDS, the Visiting Speakers Program in Oncology at McGill University and the Universite´ de Montre´al, AIM Group International, and ESMO; fees for travel, food, and beverage from Tango Therapeutics, AstraZeneca Pharmaceuticals, Amgen Inc., Guardant Health, and Dava Oncology; stock or stock options in BioNTech SE and Moderna Inc.; research grants (to institution) from Amgen Inc., Mirati Therapeutics, Boehringer Ingelheim, Merck & Co, and Novartis; Study Chair funds (to institution) from Pfizer; and research grants (spouse, to institution) from Almmune. C.M.G. reports fees for advisory committees from AstraZeneca, Bristol Myers Squibb, Jazz Pharmaceuticals, G1 therapeutics, and Monte Rosa Therapeutics, research support from AstraZeneca, and speaker's fees from AstraZeneca and Beigene. T.C. reports speaker fees/honoraria from the Society for Immunotherapy of Cancer (SITC), Bristol Myers Squibb, Roche, Medscape, IDEOlogy Health, Physicians' Education Resource®, LLC (PER®), OncLive and PeerView; travel, food and beverage expenses from Physicians' Education Resource®, LLC (PER®), Dava Oncology, SITC, International Association for the Study of Lung Cancer, IDEOlogy Health and Bristol Myers Squibb; advisory role/consulting fees from MedImmune/AstraZeneca, Bristol Myers Squibb, EMD Serono, Merck, Genentech, Arrowhead Pharmaceuticals and Regeneron; and institutional research funding from MedImmune/AstraZeneca, Bristol Myers Squibb, Boehringer Ingelheim and EMD Serono, all outside of the submitted work. S.J.G. reports research support from AstraZeneca, BMS, and Millenium Pharmaceuticals, all outside of the submitted work. P.P.L. reports personal fees from Viewray, Inc., AstraZeneca, Inc., personal fees and non-financial support from Varian, Inc., personal fees from Genentech, Inc., outside the submitted work. D.L.G. reports honoraria for scientific advisory boards from AstraZeneca, Sanofi, Alethia Biotherapeutics, Menarini, Eli Lilly, 4D Pharma and Onconova, and research support from Janssen, Takeda, Astellas, Ribon Therapeutics, NGM Biopharmaceuticals, Boehringer Ingelheim, Mirati Therapeutics and AstraZeneca, all outside of the submitted work. X.L. reports receiving consultant and advisory fee from Eli Lilly, AstraZeneca, EMD Serono, Daiishi Sanko, Spectrum Therapeutics, Boehringer Ingelheim, Hengrui Therapeutics, Novartis, and research funding from Eli Lilly, Boehringer Ingelheim, all outside of the submitted work. J.F.G. has served as a compensated consultant or

received honoraria from Bristol-Myers Squibb, Genentech/Roche, Takeda, Loxo/Lilly, Blueprint, AstraZeneca, Gilead, Moderna, AstraZeneca, Curie Therapeutics, Mirati, Nuvalent, Pfizer, Novartis, Merck, iTeos, Karyopharm, Silverback Therapeutics, and GlydeBio; research support from Novartis, Genentech/Roche, and Takeda; institutional research support from Bristol-Myers Squibb, Tesaro, Moderna, Blueprint, Jounce, Array Biopharma, Merck, Adaptimmune, Novartis, and Alexo; and has an immediate family member who is an employee with equity at Ironwood Pharmaceuticals. J.V.H. reports receiving advisory/consulting fees from Astra-Zeneca, Boehringer-Ingeheim, Catalyst, Genentech, GlaxoSmithKline, Guardant Health, Foundation Medicine, Hengrui Therapeutics, Eli Lilly, Novartis, Spectrum, Sanofi, Takeda Pharmaceuticals, Mirati Therapeutics, Bristiol-Myers Squibb, BrightPath Biotherapeutics, Janssen Global Services, Nexus Health Systems, EMD Serono, Pneuma Respiratory, Kairos Venture Investments, Leads Biolabs, RefleXion, and research funding from GlaxoSmithKline, AstraZeneca, Spectrum, all outside of the submitted work. Y.L. reports research funding from Merck, MacroGenics, Tolero Pharmaceuticals, AstraZeneca, Vaccinex, Blueprint Medicines, Harpoon Therapeutics, Sun Pharma Advanced Research, Bristol-Myers Squibb, Kyowa Pharmaceuticals, Tesaro, Bayer HealthCare, Mirati Therapeutics, Daiichi Sankyo. Scientific Advisory boards for AstraZeneca Pharmaceuticals, Janssen Pharmaceutical, Lilly Oncology, Turning point therapeutics. Consultation fee from AstraZeneca. Honorarium from Clarion Health Care. J.Z. reports grants from Merck, Novartis, Johnson and Johnson, personal fees from BMS, AZ, Novartis, Johnson and Johnson, GenePlus, Hengrui, Innovent, outside the submitted work. N.I.V. receives consulting fees from Sanofi, Regeneron, Oncocyte, and Eli Lilly, and research funding from Mirati, outside the submitted work. The other authors declare no competing interests in the submitted work.

## Additional information

[1]Department of Thoracic/Head and Neck Medical Oncology, The University of Texas MD Anderson Cancer Center, Houston, TX, USA. [2]Department of Imaging Physics, The University of Texas MD Anderson Cancer Center, Houston, TX, USA. [3]Division of Hematology and Oncology, Mayo Clinic, Jacksonville, FL, USA. [4]Department of Biostatistics, The University of Texas MD Anderson Cancer Center, Houston, TX, USA. [5]Department of Radiology, Massachusetts General Hospital, Boston, MA, USA. [6]Department of Radiation Oncology, The University of Texas MD Anderson Cancer Center, Houston, TX, USA. [7]Department of Thoracic Imaging, The University of Texas MD Anderson Cancer Center, Houston, TX, USA. [8]Department of Thoracic and Cardiovascular Surgery, The University of Texas MD Anderson Cancer Center, Houston, TX, USA. [9]Department of Pathology, The University of Texas MD Anderson Cancer Center, Houston, TX, USA. [10]Department of Medicine, Massachusetts General Hospital, Boston, MA, USA. [11]Department of Genomic Medicine, The University of Texas MD Anderson Cancer Center, Houston, TX, USA. [12]These authors contributed equally: Lingzhi Hong, Muhammad Aminu, Shenduo Li. [13]These authors jointly supervised this work: Yanyan Lou, Jia Wu, Jianjun Zhang, Natalie I. Vokes. ✉e-mail: jzhang20@mdanderson.org; nvokes@mdanderson.org

