## [Peer Review File · Nature Communications]

Efficacy and clinicogenomic correlates of response to immune checkpoint inhibitors alone or with chemotherapy in non-small cell lung cancerREVIEWER COMMENTS

Reviewer #2 (Remarks to the Author): expert in biostatistics and clinical trial analysis

This paper explores the role of combination immune checkpoint inhibitor (ICI) therapy with chemotherapy (ICI-chemo) over ICI monotherapy (ICI-mono) in non-small cell lung cancer (NSCLC) in terms of progression-free and overall survival outcomes. The authors provide a lot of valuable results and a detailed discussion. I list below my suggestions/concerns regarding the paper:

- Although the authors present many valuable insights, the statistical analysis performed to get those results is not well structured. As it is written, the methods section gives the impression that there was no clear direction in the statistical analysis plan. There are several examples of so many relationships in so many directions explored in this paper that the statistical analysis plan reads as an ad-hoc mechanism. Researchers might have several questions to be explored from the same data set, but from a statistical point of view, we would not necessarily fit a different model to each of those questions. For instance, first, a Cox model was fit to identify the effects of covariates on survival outcomes; second, another Cox model was fit to survival outcomes after propensity score matching; a third Cox model fit to see the effects of genomic markers, and finally, a fourth Cox model was fit to see the interaction between genetic markers and treatments. Furthermore, one logistic regression was fit to see the association between genomic features and 3-month progression; another model was fit to see the interaction between genomic features and treatment; the latter will already give information on the former. The methods section gives the impression that the authors performed a lot of analysis and explored a lot of associations, but the analysis seems quite repetitive (also leading to questions of an increase in type I error rate) and not very well planned. I suggest the authors reconsider their statistical plan, write a more structured plan that removes the repetitions in modeling.
- Another issue is the univariate analysis that was performed, particularly the analysis for PD-L1 expression, which ignores the effect of other covariates. Univariate analysis can be informative, but in this type of analysis, the significance of a covariate might be due to not controlling for other covariates. Therefore, it is advisable to follow a univariate analysis with an analysis that controls for other covariates.
- It is not clear whether possible moderation effects are considered in all models, that is, whether possible interactions among the covariates were explored in all models.
- Minor comment: I suggest making sure all the abbreviations are explained the first time they appear in the manuscript, e.g., the abbreviations PFS and OS appear in the results section; however, their definitions were given later in the paper (in the methods section).

Reviewer #3 (Remarks to the Author): expertise in immune checkpoint blockade in NSCLC

In this retrospective study of 1,139 NSCLC patients, treatment with ICI-mono vs ICI-chemo was associated with PFS, OS, genetic and clinicopathological markers. Several clinically relevant associations were defined in this study, although many have been previously shown, that could help define patient populations who may specifically benefit from ICI-chemo vs. ICI alone. A notable finding was that while short term PFS is increased by ICI-chemo, long-term PFS and OS is not

different between ICI-mono vs ICI-chemo. A finding of clinical significance is that in addition to low PD-L1, alterations in cell cycle genes, STK11 loss and greater disease burden are predictors higher risk of early progression and therefore these patients may benefit more from ICI-chemo than ICI alone. The findings are tempered by the authors clear mention of study limitations and the fact that these are meant to be exploratory findings that need to be validated before changes in patient care are considered. An issue of note is that many of the primary findings have been previously reported including the negative effect of STK11 alterations with ICI response, but nonetheless this large cohort investigation of multiple key parameters in a single study has obvious and important implications for patient care.

Minor concerns: the authors should specify in the results that this study is on patients at a single institution. Also, the authors should specify for the general reader the nature of mutations of interest, e.g., inactivating for STK11 and potentially JAK2 vs. activating mutations.

Reviewer #4 (Remarks to the Author): clinical expert in NSCLC

Thank you for the opportunity to review this article “Efficacy and clinicogenomic correlates of response to immune checkpoint inhibitors alone or with chemotherapy in non-small cell lung cancer”. This is a retrospective, single institution study which highlights importance of identifying patients with stage IV NSCLC at risk of early progression of their disease. The authors identify both radiologic and histological factors, including PD-L1 score, liver metastases, and alterations in STK11, JAK2, and CDK6 as factors associated with a higher risk of early progression of disease on immune checkpoint inhibitor monotherapy. They also identify no significant difference in long-term outcomes in patients who received sequential versus concurrent immunotherapy and chemotherapy, which has recently been a growth area in lung cancer research.

Major Comments

1. Methods: 459 pts were treated with 2nd line IO or ICI-chemo. Is this truly representative of the population in which these treatment decisions are being made, as the choice between 1ci and 1ci-chemo is now exclusively first line?
2. Methods: PD-L1 proportion was high ($\geq 50\%$), low (1-49%), negative, and unknown in 243 (21.3%), 277 (24.3%), 255 (22.4%), and 364 (31.9%) patients, respectively. Suggest ‘low’ here should be renamed as ‘intermediate.’
3. Methods: Immortal time bias. The authors included all patients who had ≥ 2 cycles of treatment, did the number of cycles given have an effect on the clinical endpoints measured? Was the study powered to detect differences in outcomes with each immune checkpoint inhibitor combination given?
4. Results: the differential outcome by histology was not seen in prospective studies. Would the authors like to comment on this in the discussion

5. Results: suggest adding details on how the propensity score adjustment was performed, in the methods section of the MS.

6. Results: Can the authors explain why the KRA-mt pts treated with ICI mono had a longer PFS in this dataset.

7. Results: In the ICI-mono patients, liver metastases, metastatic stage (IVa vs IVb/c),

PD-L1, STK11, JAK2, and CDK6 alterations were the top features associated with 3-

month progression. Suggest add PD-L1 cut-off in this statement and how PD-L1 was analyzed.

8. Results: The authors identify that upfront combination treatment, and sequential ICI and chemotherapy is associated with improved 3 month PFS but there is no difference in long term outcomes. did the authors investigate the toxicities/tolerability of in both treatment strategies? Combination treatment has previously been shown to have a higher incidence of treatment related toxicity and impact on patients quality of life. Was the performance status of these patients included as a covariate in this analysis?

Minor Comments

- In 3 sections the authors switch between using the terms ICI-chemo, chemoimmunotherapy and IO-chemo
- Figure S1 shows that those with other types of NSCLC were excluded, but in table 1 there are 64 patients with other types of NSCLC. Did the authors detect any additional clinico-genomic predictors in this group?
- Although tumour mutational burden is missing from this analysis which has been shown to correlate with response to immune checkpoint blockade, this is acknowledged by the authors in the discussion, and their use of smoking status as a clinical proxy. This could potentially be explored in further studies.

Reviewer #5 (Remarks to the Author): expert in lung cancer genomics

The authors examined real-world lung cancer patient data (n=1,139) who received ICI-chemo or ICI-mono therapy in routine cancer treatment to challenge the following questions (as described by the authors in the introduction), which to date have not been adequately answered.

1. Who should receive combination therapy vs ICI monotherapy (ICI-mono)?
2. Clinicogenomic predictors of ICI response
3. Whether the addition of chemotherapy affects short- and long-term outcomes to ICI therapy in NSCLC?

For the issue 2, several gene alterations were picked up in the first-line cases (n=680): STK11, JAK2, and CDK6 with early progression with ICI-mono, and CDKN2A and CCND1 with worse long-term outcome with ICI-chemo.

In Task 3, ICI-mono was found to have a higher rate of progression at 3 and 6 months compared to ICI-chemo, but the long-term progression-free survival and overall survival were comparable. Based on these findings, the authors discuss that biomarkers such as STK11, JAK2, and CDK6 might be useful for identifying patients who may benefit from ICI-chemotherapy.

These findings are very interesting and may lead to a solution to Issue 1. This is a very interesting story based on real world data. However, STK11 seems to be associated with 3-month progression in both ICI-chemo and ICI-mono (Fig. 3A, B). Thus, the genomic predictors of Issue 1 are still not firmly determined.

They also found that the long-term prognosis is comparable between sequential and concurrent ICI and chemotherapy. This is interesting data, but needs to be validated in several cohorts before conclusions can be drawn.

In summary, the authors present some very interesting issues based on real-world data. However, they are not robust enough and do not have a strong enough impact to change routine oncology.

RESPONSE TO REVIEWER COMMENTS

Reviewer #2 (Remarks to the Author): expert in biostatistics and clinical trial analysis

This paper explores the role of combination immune checkpoint inhibitor (ICI) therapy with chemotherapy (ICI-chemo) over ICI monotherapy (ICI-mono) in non-small cell lung cancer (NSCLC) in terms of progression-free and overall survival outcomes. The authors provide a lot of valuable results and a detailed discussion. I list below my suggestions/concerns regarding the paper:

1. Although the authors present many valuable insights, the statistical analysis performed to get those results is not well structured. As it is written, the methods section gives the impression that there was no clear direction in the statistical analysis plan. There are several examples of so many relationships in so many directions explored in this paper that the statistical analysis plan reads as an ad-hoc mechanism. Researchers might have several questions to be explored from the same data set, but from a statistical point of view, we would not necessarily fit a different model to each of those questions. For instance, first, a Cox model was fit to identify the effects of covariates on survival outcomes; second, another Cox model was fit to survival outcomes after propensity score matching; a third Cox model fit to see the effects of genomic markers, and finally, a fourth Cox model was fit to see the interaction between genetic markers and treatments. Furthermore, one logistic regression was fit to see the association between genomic features and 3-month progression; another model was fit to see the interaction between genomic features and treatment; the latter will already give information on the former. The methods section gives the impression that the authors performed a lot of analysis and explored a lot of associations, but the analysis seems quite repetitive (also leading to questions of an increase in type I error rate) and not very well planned. I suggest the authors reconsider their statistical plan, write a more structured plan that removes the repetitions in modeling.

Author response: Thank you for this constructive comment. We agree that the structure and direction of our statistical analyses were not presented clearly in the methods section, and we have updated our statistical methods section to more clearly delineate the primary and exploratory analyses of our study. The relevant portion now reads as follows:

The primary outcome analysis focused on comparing treatment outcomes in patients treated with first-line ICI-mono vs ICI-chemo, with planned subgroup analysis by PD-L1 grouping and smoking status¹. Survival estimates were obtained using the Kaplan-Meier method and crude

differences between groups were assessed through the log-rank test. To assess whether the effect of treatment changed over time, Aalen's additive hazards model was applied². To address possible systemic differences between the ICI-mono vs ICI-chemo group and reduce potential bias from these confounding factors, a propensity score-based analysis was performed. Propensity scores were estimated by regressing treatment assignment (ICI-mono vs ICI-chemo) on key prognostic variables (age, sex, tobacco use, histology, metastatic stage at ICI initiation, brain metastasis, liver metastasis, and PD-L1), and applied using the inverse probability of treatment weighting methodology (IPTW). Differences in primary outcomes between treatment groups were assessed in the IPTW-adjusted cohort. Post-weighting balance in covariates between treatment groups was evaluated using the standardized mean difference (SMD) approach, with an imbalance defined as SMD >0.1.

In addition to the primary outcome analysis, focused exploratory analyses were performed to 1) identify any clinicogenomic features that associate with ICI outcome, and 2) determine whether any of these features were differentially predictive in the ICI-mono vs ICI-chemo context. Accordingly, Cox proportional hazards regression models were applied to identify clinicogenomic variables associated with PFS and OS in the ICI-mono and ICI-chemo contexts, and logistic regression model was applied to identify clinicogenomic variables associated with 3-month PFS. Clinical features were selected for multivariate analysis if they were significant on univariate analysis for PFS and OS, and pair-wise interaction terms that were significant were included in the multivariate model. Those variables that were significant (p-value <0.05) on these regression analyses were selected for subgroup analysis to further interrogate the treatment effect. Exploratory subgroup and sensitivity analyses were not adjusted for multiple comparisons. The R package subtee³ was used to generate treatment effect estimates.

We note that we worked closely with our statistical co-authors (Dr. J. Jack Lee, Dr. Xuetao Lu) to formulate this analytic approach and construct appropriate statistical models. In keeping with this methodologic structure, we have re-ordered our figures, so that Figures 1-2 reflect the primary assessment of outcome, and the subsequent figures reflect the results of our exploratory analyses. We also believe this structure better highlights the two complementary and non-overlapping goals in the exploratory analyses, which is to first determine whether any clinicogenomic features associate with outcome, and then secondly, determine whether these associations are treatment specific. To avoid excessive subgroup testing and type I error, we use the first analysis to nominate features of interest, and then perform focused assessment for treatment effect in these subgroups. We also excluded clinical variables from multivariate analysis that were not significant (gender, brain metastases) to avoid overfitting and analysis of too many variables. This overall analytic structure is reflected in this table:

	Methods	Goals/Questions
Primary analysis	Kaplan-Meier + Log-rank test Propensity score adjustment	Do the primary (PFS, OS) and secondary outcomes (3-mo PFS) differ between ICI-mono vs ICI-chemo?
Secondary analyses	Cox PHM (PFS/OS) Logistic regression (3-mo PFS)	Identify clinicogenomic features that associate with outcome
	Subgroup treatment effect analysis	Determine whether subgroups (defined by above) have a significantly different ICI-mono vs ICI-chemo treatment effect

Reviewer Figure 1. Schematic of statistical analysis structure.

- Another issue is the univariate analysis that was performed, particularly the analysis for PD-L1 expression, which ignores the effect of other covariates. Univariate analysis can be informative, but in this type of analysis, the significance of a covariate might be due to not controlling for other covariates. Therefore, it is advisable to follow a univariate analysis with an analysis that controls for other covariates.

Author response: Thank you for this comment; we agree that multivariate adjustment is important, and while we had performed multivariate adjustment in our other analyses, we had neglected to do so in the analysis of PD-L1. Thank you for bringing this oversight to our attention. We have updated our approach so that we first perform univariate analysis of clinicogenomic features associated with PD-L1, followed by multivariate adjustment of the features with p-value < 0.05 on univariate analysis. We found that the strongest associations remained significant after multivariate adjustment, and only MET alterations lost significance under this analysis:

Reviewer Figure 2: Univariate (left panel) and multivariate (right panel) association of clinicogenomic features with PD-L1 expression.

These results are included as **Supplementary 4**, and the methods have been updated to reflect this approach as well.

3. It is not clear whether possible moderation effects are considered in all models, that is, whether possible interactions among the covariates were explored in all models.

Author response: Thank you for highlighting this point. Our analysis focused primarily on the interaction between treatment (ICI-monotherapy vs ICI-chemotherapy) and outcome, with secondary analyses to study the association between clinicogenomic variable, outcome, and treatment effect, and we have clarified our methods to explicitly highlight these analytic goals. We note that it is not possible to explore the interactions between all clinicogenomic variables (> 70 features considered), but we agree that clinicogenomic features may have significant interrelationships, some of which we in fact highlight in the above analysis of PD-L1. In our more focused analysis of the clinical variables and treatment outcome in the whole cohort (Fig 3), we have updated our statistical approach to assess pair-wise interactions between the clinical variables that were significant on univariate analysis. This analysis identified a significant interaction effect between PD-L1 expression <50% and metastatic stage ($p < 0.0418$), and we therefore included this interaction term in our multivariate outcome analyses. The updated statistical methods section copied above includes our analysis of interaction-terms.

For the integrative clinicogenomic analysis of 3-month progression, we used a feature selection strategy to avoid redundant or co-linear feature inclusion; this allowed us to build a more parsimonious model while also providing some biological insight as to which features are the most predictive of 3-month progression in the ICI-mono and ICI-chemo contexts.

4. Minor comment: I suggest making sure all the abbreviations are explained the first time they appear in the manuscript, e.g., the abbreviations PFS and OS appear in the results section; however, their definitions were given later in the paper (in the methods section).

Author response: We thank the reviewer for this suggestion. We have revised the text accordingly.

Reviewer #3 (Remarks to the Author): expertise in immune checkpoint blockade in NSCLC

In this retrospective study of 1,139 NSCLC patients, treatment with ICI-mono vs ICI-chemo was associated with PFS, OS, genetic and clinicopathological markers. Several clinically relevant associations were defined in this study, although many have been previously shown, that could help define patient populations who may specifically benefit from ICI-chemo vs. ICI alone. A notable finding was that while short term PFS is increased by ICI-chemo, long-term PFS and OS is not different between ICI-mono vs ICI-chemo. A finding of clinical significance is that in addition to low PD-L1, alterations in cell cycle genes, STK11 loss and greater disease burden are predictors higher risk of early progression and therefore these patients may benefit more from ICI-chemo than ICI alone. The findings are tempered by the authors clear mention of study limitations and the fact that these are meant to be exploratory findings that need to be validated before changes in patient care are considered. An issue of note is that many of the primary findings have been previously reported including the negative effect of STK11 alternations with ICI response, but nonetheless this large cohort investigation of multiple key parameters in a single study has obvious and important implications for patient care.

Author response: Thank you for your comments and thoughtful review. We agree that, within the limitations of a retrospective analysis, our analyses provide unique insights into the role of chemotherapy in ICI-treated patients, and it is one of the first that integrates and nominates clinicogenomic features that may influence the decision on whether or not to include chemotherapy with ICI. In a therapeutic landscape in which providers no longer decide whether or not to give ICIs but rather whether or not to give chemotherapy with ICIs, we believe this approach makes our results highly clinically relevant.

We note that, while several of these features have indeed been identified in previous analyses, this is the first analysis to systematically dissect and rank how these features should influence the specific clinical choice between ICI-mono and ICI-chemo; the re-discovery of several previously identified features provides corroboration that our findings are generalizable outside our datasets. We

also note several features that distinguish these findings from previously published papers: first, the association between *CDKN2A* and differential outcomes in ICI-mono vs ICI-chemo-treated patients is novel; given its validation in multiple datasets, we believe it is hypothesis generating, though certainly in need of more validation. Second, while many studies have included clinical or genomic features, there are few datasets of this size that integrate both. Importantly, our annotations include markers of disease burden and distribution (liver metastases), which are often missing from published datasets but are clinically readily available and turn out to be quite predictive. Finally, our analysis of concurrent vs sequential outcomes is novel, and lends support to the hypothesis that ICI and chemotherapy are additive rather than synergistic. We note that in this revised manuscript we have expanded on these findings and validated them in multiple independent datasets, enhancing the overall robustness of our study. While ultimately exploratory, we believe these results offer a valuable contribution to the literature and provide data that can inform the clinical questions providers are currently making in a much more data-free manner.

1. Minor concerns: the authors should specify in the results that this study is on patients at a single institution.

Author response: We thank the reviewer for this suggestion. We have clarified in our methods the institutions from which our cohorts derive, and have included this information in the figure captions as appropriate. In our revised version, we do have an external validation dataset from Mayo, and an additional more focused validation dataset from MGH specific for the question of overall survival with sequential vs concurrent ICI and chemotherapy, as requested by reviewer #5. Our updated methods are copied here:

Study population

MDACC Cohorts:

MDACC-Primary Cohort: We queried GEMINI, a University of Texas MD Anderson Cancer Center Lung Cancer Moon Shot funded internal database to identify patients treated with ICI who met the following criteria: 1) diagnosis of pathologically confirmed non-small cell lung cancer (NSCLC), including adenocarcinoma, squamous cell carcinoma, adenosquamous carcinoma, and NSCLC-not otherwise specified (NOS); 2) stage IV disease at the time of immune checkpoint inhibitor (ICI) start; and 3) received ≥ 2 cycles of ICI alone or with chemotherapy. Consistent with the FDA approvals, patients with known targetable EGFR or ALK alterations were excluded from the analysis. Clinical data were collected until September 10, 2020, when the dataset was locked for clinical outcome analysis.

MDACC-Validation Cohort: patients enrolled in GEMINI between March 2020-January 2022 (after the data lock for the MDACC-primary cohort) who were treated with ICI in the first-line setting who otherwise met the same inclusion criteria above were included in a temporally distinct validation cohort.

This study was approved by the institutional review board at MD Anderson Cancer Center

Mayo Cohort: an external validation cohort of patients treated with ICIs at Mayo Clinic was identified. The same inclusion criteria were applied to the external validation cohort from Mayo Clinic. This study was approved by the institutional review board at Mayo Clinic and all patients provided written informed consent.

MGH Cohort: an additional validation cohort for the focused question of whether overall survival differed by treatment with concurrent vs sequential ICI and chemotherapy was obtained from Massachusetts General Hospital (MGH). This cohort consisted of patients with NSCLC treated with first-line ICIs at MGH between 01/2013-01/2020. As this cohort was assembled for radiographic analyses, patients without pre-treatment chest CTs or known ILD were (n=8) were excluded.

- Also, the authors should specify for the general reader the nature of mutations of interest, e.g., inactivating for *STK11* and potentially *JAK2* vs. activating mutations.

Author response: Thank you for raising this point. We have included a co-mut plot as Figure 5C to help illustrate the relevant mutations, which consist primarily in missense/truncating mutations for *STK11*, *JAK2* activating mutations, copy number amplifications of the cell cycle genes (*CCND1*, *CCNE1*, *CDK6*, *CDK4*), and a mix of missense/truncating mutations and copy number loss of *CDKN2A* (Figure 5D). We will also make the genomic data available to the reader on publication. These figures are copied below:

Reviewer Figure 3. Figures 5C and 5D from the revised manuscript, containing a co-mutation plot from the MDACC-primary cohort of alterations in the genes significantly associated with outcome (5C), and the specific alterations in *CDKN2A* in the primary cohort and two validation cohorts (5D).

Reviewer #4 (Remarks to the Author): clinical expert in NSCLC

Thank you for the opportunity to review this article “Efficacy and clinicogenomic correlates of response to immune checkpoint inhibitors alone or with chemotherapy in non-small cell lung cancer”. This is a retrospective, single institution study which highlights importance of identifying patients with stage IV NSCLC at risk of early progression of their disease. The authors identify both radiologic and histological factors, including PD-L1 score, liver metastases, and alterations in *STK11*, *JAK2*, and *CDK6* as factors associated with a higher risk of early progression of disease on immune checkpoint inhibitor monotherapy. They also identify no significant difference in long-term outcomes in patients who received sequential versus concurrent immunotherapy and chemotherapy, which has recently been a growth area in lung cancer research.

Major Comments

- Methods: 459 pts were treated with 2nd line IO or ICI-chemo. Is this truly representative of the

population in which these treatment decisions are being made, as the choice between 1ci and 1ci-chemo is now exclusively first line?

Author response: We agree that this important point. It is certainly the case that the landscape of NSCLC therapy evolved during the time that patients in our cohort were being treated, and that ICI therapy moved from the chemo-refractory context to the front-line. While this does add an element of heterogeneity to our overall cohort, we note several points that we believe addresses the reviewer's concern about the validity of our analysis in this now-changed treatment landscape:

- 1) We agree with the reviewer that survival outcomes may differ by treatment context; we therefore performed the primary outcome analysis in just the first-line treated patients, which represents the majority of our primary cohort (60%). Additionally, our validation cohorts (Mayo and MDACC-validation) were exclusively first-line patients (n=482 in total). Therefore, the bulk of our analyzed patient population reflect current clinical practice, and the primary analysis was performed in the specific population of interest.
- 2) For the second set of analyses, these focused less on comparing treatment-specific outcomes and more on understanding how clinicogenomic features associate with outcome. The growing body of evidence suggests that the predictors of benefit to ICI are not specific to line of therapy, e.g. PD-L1 and TMB have been predictive in both chemo-refractory and first-line cohorts. It is therefore reasonable to assess these features in a mixed cohort, However, we did include line of therapy as a covariate in the clinical analyses and noted that line of therapy was not significantly associated with outcome on multivariate analysis. To address the reviewer's concern more explicitly, we also repeated the analysis of clinical predictors of outcome in the first-line cohort alone and note a nearly identical set of associations, confirming that the predictors of ICI outcome are not likely to be contingent on line of therapy. These analyses are now included as Supplementary Figure 13, and are copied here below:

Reviewer Figure 4. Supplementary Figure 13. Hazard ratios depicting the association between clinical features and progression-free survival (PFS) (A) and overall survival (OS) (B) in ICI-mono (left panel) and ICI-chemo (right panel). Values from multivariate analysis are shown in pink. Forest plot depicts hazard ratio with 95% confidence interval.

Given these considerations, we felt that analysis of clinicogenomic predictors in the whole cohort was reasonable, and we adopted this approach to maintain consistency while preserving as much power as possible for the genomic analyses⁴. The validation of these findings in our first-line validation cohorts also supports that these findings are applicable to the current treatment context. Ultimately, the results of our analyses will need to be validated in other datasets, and we anticipate that most ICI analyses going forward will be almost exclusively in first-line patients.

3) Finally, though this was not the driving reason for including these patients, we do note that there

may be some advantages to including patients treated in the second-line or higher in addition to simply increasing our power. First, in contrast to the newer cohorts, these patients have ample follow-up time with more progression and mortality events; and second, these patients increase our cohort of patients with PD-L1 < 50% treated with ICI-monotherapy, allowing us to study clinicogenomic predictors of benefit in different treatment contexts in a manner that is less confounded by PD-L1.

2. Methods: PD-L1 proportion was high ($\geq 50\%$), low (1-49%), negative, and unknown in 243 (21.3%), 277 (24.3%), 255 (22.4%), and 364 (31.9%) patients, respectively. Suggest 'low' here should be renamed as 'intermediate.'

Author response: We thank the reviewer for this suggestion. We revised the PD-L1 expression level as high ($\geq 50\%$), intermediate (1%-49%), and low (< 1%).

3. Methods: Immortal time bias. The authors included all patients who had ≥ 2 cycles of treatment, did the number of cycles given have an effect on the clinical endpoints measured? Was the study powered to detect differences in outcomes with each immune checkpoint inhibitor combination given?

Author response: Thank you for raising these important questions. While the restriction to patients who received ≥ 2 cycles of treatment is necessary to ensure adequate follow-up for outcome assessment, it does potentially exclude patients with more rapid clinical deterioration. We feel that this restriction is reasonable, as it excludes those patients who were likely too sick to receive any benefit from therapy and may therefore confound our analysis of therapy-specific predictors. We note that other retrospective analyses took a similar approach by excluding those patients who were not radiographically evaluable^{5,6}. In total, only 51 patients were excluded for stopping therapy after one cycle, which is not likely to have a significant effect.

The broader question that the reviewer raises about the association between number of cycles and clinical endpoints is important but very difficult to disentangle, as patients who are benefitting from a therapy are likely to continue to receive that therapy. Conversely, and reflecting complexity of this question is in the context of immunotherapies, ICI-treated patients with immune-related toxicities may stop therapy but continue to benefit from preserved disease control. Assessing the association between number of cycles of therapy and outcome is therefore a very interesting question that would require more detailed information about reasons for stopping therapy, and is therefore beyond the scope of this analysis. We did, however, perform detailed clinical annotations to ensure that our outcome measure (PFS) reflects disease progression, whereas other retrospective studies have used time on therapy as a surrogate which may not be valid in the context of ICIs for the reasons noted above.

With respect to ICI combination regimens, in the first-line setting, 96.3% (362/376) were treated with pembrolizumab and a histology-guided chemotherapy regimen: 314 with pemetrexed-platinum

combination, 48 with taxane-platinum combination. Comparisons of PFS and OS between these two groups were presented below.

Reviewer Figure 5. Progression-free and overall survival in first-line patients treated with ICI-chemo, stratified chemotherapy regimen.

While it would be interesting to dissect the efficacy of different ICI and chemotherapy regimens, as the reviewer suggests we are not powered for this analysis, and these outcomes are likely heavily confounded by the factors underlying treatment regimen selection, including histology and treatment context.

4. Results: the differential outcome by histology was not seen in prospective studies. Would the authors like to comment on this in the discussion

Author response: We agree that this was an interesting finding. However, when we expanded our analysis to include validation cohorts, we did not see similarly striking differences in PFS by histology in the MDACC validation cohort; in this cohort there was also a much clearer association between PD-L1 and outcome in squamous cell carcinoma patients. In both cohorts, there was a clear difference in OS in patients with adenocarcinoma compared to squamous cell carcinomas.

Reviewer Figure 6: Manuscript Supplementary Figure 15. A-B) Kaplan-meier plot for progression-free survival (PFS) stratified by histology in A) MDACC-primary cohort, B) MDACC-validation cohort. C-D) Kaplan-meier plot for overall survival (OS) stratified by histology in C) MDACC-primary cohort and D) MDACC-validation cohort. E-F) Kaplan-meier plot for PFS in patients with squamous cell carcinoma histology, stratified by treatment regimen, in E) MDACC-primary cohort and F) MDACC-validation cohort.

Given that our validation cohort did not support a clear difference in PFS, we have de-emphasized these findings in our revised manuscript, highlighting merely the outcome associations described above which

are included as **Supplementary Figure 15**. We do note as a point of interest, however, that squamous histology and histology-specific ICI response predictors are relatively understudied, as most cohorts are either dominated by adenocarcinoma or explicitly exclude squamous histology. There have also been reports supporting a weaker association between PD-L1 and outcome in squamous cell carcinomas⁷, suggesting that the predictive association may be affected by other unknown factors. Further histology-specific analyses in larger cohorts may yield interesting insights in the future as to differential predictors of response in NSCLC histologies.

5. Results: suggest adding details on how the propensity score adjustment was performed, in the methods section of the MS.

Author response: We thank the reviewer for this suggestion. We have added the following description of our propensity score adjustment to the methods:

Propensity scores were estimated by regressing treatment assignment (ICI-mono vs ICI-chemo) on key prognostic variables (age, sex, tobacco use, histology, metastatic stage at ICI initiation, brain metastasis, liver metastasis, and PD-L1), and applied using the inverse probability of treatment weighting methodology (IPTW). Differences in primary outcomes between treatment groups were assessed in the IPTW-adjusted cohort. Post-weighting balance in covariates between treatment groups was evaluated using the standardized mean difference (SMD) approach, with an imbalance defined as SMD >0.1.

6. Results: Can the authors explain why the KRA-mt pts treated with ICI mono had a longer PFS in this dataset.

Author response: Thank you for highlighting this result. *KRAS* mutant NSCLC is clinically and genomically heterogeneous, with features that predispose both to ICI response (e.g. association with smoking, higher TMB, and often higher PD-L1⁸⁻¹⁰) and to ICI resistance (e.g. *STK11/KEAP1* co-mutations¹¹).

Consequently, the published clinical data is also somewhat heterogeneous; however, most studies have demonstrated better or at least comparable outcomes in patients with *KRAS* mutations compared to those without oncogenic driver alterations, which is in contrast to other oncogenic drivers alterations which consistently associate with worse outcomes¹²⁻¹⁴. However, due to these complex inter-related features, the association between *KRAS* mutation and efficacy likely needs to be disentangled in the context of clinical variables along with genomic features (*TP53*, *KEAP1*, *STK11*, and TMB). In response to the reviewer's query, we analyzed the *KRAS*-mutant subcohort and noted that *KRAS* mutations associated with former smoking status and adenocarcinoma histology, along with a trend toward higher PD-L1 expression (Reviewer Figure 7A).

A

Parameters, n (%)	KRAS negative (n=480)	KRAS positive (n=255)	P value
Age at ICI started			
18-65	244 (68)	117 (32)	0.201
> 65	236 (63)	138 (37)	
Gender			
Male	293 (73)	110 (27)	<0.001
Female	187 (56)	145 (44)	
Tobacco use			
Never	118 (83)	24 (17)	<0.001
Former	277 (59)	191 (41)	
Current	85 (68)	40 (32)	
Histology			
LUAD	364 (61)	229 (39)	<0.001
LUSC	94 (90)	10 (10)	
Others	22 (58)	16 (42)	
PD-L1 expression			
< 1%	134 (71)	54 (29)	0.075*
1-49%	128 (68)	61 (32)	
≥50%	99 (60)	66 (40)	
Unknown	119 (62)	74 (38)	
Treatment			
ICI-mono	314 (67)	152 (33)	0.120
ICI-chemo	166 (62)	103 (38)	
Line of ICI			
First line	279 (63)	167 (37)	0.060
Second line	157 (68)	75 (32)	
> 2 nd line	44	13	
Metastatic status at ICI start			
IVA	199 (66)	104 (34)	0.860
IVB	281 (65)	151 (35)	
Liver metastasis at ICI start			
No	417 (66)	215 (34)	0.341
Yes	63 (61)	40 (39)	
Brain metastasis at ICI start			
No	339 (65)	183 (35)	0.746
Yes	141 (66)	72 (34)	

B

C

Reviewer Figure 7. A) Table of clinical features in *KRAS* mutant negative vs *KRAS* mutant positive patients. B-C) Progression-free survival stratified by *KRAS* and *STK11* mutation status in B) the MDACC-primary cohort and C) the MDACC validation cohort.

We also note a high number of patients without *STK11* co-mutations (21.6% *STK11* co-mutated, 57/264), with even better outcomes in the *KRAS*-mutant/*STK11*-wildtype population (Reviewer Figure 7B).

Therefore, we suspect that at least in our primary cohort, the *KRAS* mutant patients overlapped with a number of favorable features, and we may have also captured the subgroup of *KRAS* patients that have very good ICI responses. Notably, in the MDACC validation cohort, while *STK11*-mutated/*KRAS*-mutated patients clearly had the worst PFS, the preferential benefit in the other *KRAS* mutant patients is not as clear, highlighting the clinical variability that can occur in this heterogeneous patient subgroup (Reviewer Figure 7C). To fully dissect these relationships we would need more complete genomic data (KEAP1 and

TMB are missing from our dataset), like from an even larger, multi-institutional effort, and given the complexity and multifactorial nature of the problem, along with multiple published reports in this space, we chose not to discuss these findings in detail in this analysis. However, we have included the above figure as Supplementary Figure 20, and added the following section to our results, copied here:

KRAS alterations, which were one of the few genomic associations with improved PFS, also associated with a history of tobacco exposure, adenocarcinoma histology, and trended toward higher PD-L1 expression in the MDACC-primary cohort, consistent with a more ICI-responsive clinical phenotype (Supplementary Fig 20A). However, KRAS mutations were not strongly associated with improved PFS in the MDACC validation cohort (HR 0.84, 95% CI 0.45-1.6, P=0.17), reflecting the heterogeneity of this NSCLC subgroup²⁷⁻³⁰, though STK11 co-mutation associated with worse outcomes in both cohorts, as previously reported (Supplementary Fig 20B-C)³¹.

7. Results: In the ICI-mono patients, liver metastases, metastatic stage (IVa vs IVb/c), PD-L1, STK11, JAK2, and CDK6 alterations were the top features associated with 3-month progression. Suggest add PD-L1 cut-off in this statement and how PD-L1 was analyzed.

Author response: Thank you for pointing out this ambiguity. In our methods, we describe the process of PD-L1 staining and the assigned classification of PD-L1 <1% as low, PD-L1 between 1 and 50% as intermediate, and PD-L1 > 50% as high. We have also edited our figure depicting the rank order of predictive features (Figure 6A and 6D) so that it now clarifies specifies the PD-L1 levels that achieve significance, and we have colored the bar plot to indicate whether the features have a positive or negative association with PFS, making it more visually obvious that low PD-L1 (<1%) and high PD-L1 (>50%) are the PD-L1 levels that associate most strongly with 3-month progression-free survival in opposing directions. We believe that this edited figure is easier to interpret and appreciate the reviewer's suggestion. This updated figure is copied below. (We also note that the ranked feature list has changed very slightly in this revised version because we edited the input list of features in the MDACC-primary cohort to include those variables present in the MDACC cohorts and the Mayo validation cohort, thereby expanding the generalizability of the model. We still performed all of the model training and cross-validation within the MDACC-primary cohort, and used the Mayo and MDACC-validation cohorts as independent validation datasets).

Reviewer Figure 8. Figures 6A and 6D from the manuscript depicting the contributed of ranked features for prediction of 3-month PFS in ICI-mono (A) and ICI-chemo (D).

8. Results: The authors identify that upfront combination treatment, and sequential ICI and chemotherapy is associated with improved 3 month PFS but there is no difference in long term outcomes. did the authors investigate the toxicities/tolerability of in both treatment strategies? Combination treatment has previously been shown to have a higher incidence of treatment related toxicity and impact on patients quality of life. Was the performance status of these patients included as a covariate in this analysis?

Author response: This is an important point. We do not discuss tolerability/toxicity in detail in the manuscript because we do not have this data for our patient cohorts, which is not feasible in a real-world retrospective analysis of this size nor are such data uniformly captured in the medical record. As patient-reported outcome data becomes more standardized and routinely collected, this will make these questions more tractable and will add an important dimension to similar retrospective analyses. While we cannot specifically disentangle the competing effects of toxicity vs disease control on the survival outcomes we examined, our outcome analyses would reflect whether chemotherapy is negatively affecting survival outcomes due to increased toxicities over and above what it offers in terms of disease control. The real-world balance in these intrinsic trade-offs that providers themselves would be negotiating is therefore reflected in our data. That being said, we also note that our findings do have important implications that bear directly on this question. While we focus to some extent on highlighting the features of those patients who may need chemotherapy with ICIs, the converse implication of our results is that there may be many patients who can safely be treated with ICI-monotherapy to start.

Specifically, the absence of compelling evidence for synergy between chemotherapy and ICIs suggests that providers can safely use ICI-monotherapy to start in patients with a favorable risk profile who are likely to be salvageable with sequential chemotherapy if their disease progresses. We have edited our discussion to highlight this point more explicitly, as follows, with the relevant addition bolded below:

*Importantly, our results also suggest that chemotherapy does not synergistically increase the likelihood of long-term benefit. This interpretation is supported by the observation that many of the features that predict long-term PFS to ICI-mono, such as PD-L1, are the same as those that predict long-term PFS to ICI-chemo, indicating that patients who experience long-term benefits to ICIs do so because of their particular tumor-immune features rather than from the inclusion of chemotherapy. The observation that patients treated with ICIs and chemotherapy in sequence rather than simultaneously have comparable OS is further data in support of this interpretation. Consistent with our data, simulation analyses have shown that chemotherapy added to ICIs improves outcomes by combining therapies with non-overlapping populations of responders, consistent with an additive rather than synergistic benefit^{32,33}. **These findings have important therapeutic implications in that they suggest that patients without the high-risk features described above, or those who may be reasonably likely to achieve a second line of therapy, can be safely treated with upfront ICI-monotherapy and thereby avoid the toxicities of combination therapy. Whether the absence of benefit from chemotherapy over time arises in part from increased treatment-related toxicities cannot be determined from our data, but is an additional variable that will need to be explored as more real-world toxicity data becomes available.***

Regarding the point about performance status, we did not include performance status in our analyses because ECOG assessments from the medical records were imprecise and somewhat variable; the vast majority of patients (70.9%) were given an ECOG of 0-1, and another 20% were missing this data all together. Given the minimal additional information that reported ECOG was providing, we omitted this variable from our analysis so as to not lose more patients due to incomplete data.

ECOG score for patients treated with ICI as first line therapy.

ECOG score	Total	ICI mono	ICI - combine	P*
All	680	304 (44.7)	376 (55.3)	0.024
0-1	482 (70.9)	190 (62.5)	292 (77.7)	
2-3	58 (8.5)	32 (10.5)	26 (6.9)	
unknown	140 (20.6)	82 (27.0)	58 (15.4)	

We note that the slight imbalance in ECOG scores would tend to skew the data in favor of ICI-chemo, suggesting that our conclusion that ICI-mono offers equivalent long-term outcomes is not likely affected by this potential imbalance.

Minor Comments

1. In 3 sections the authors switch between using the terms ICI-chemo, chemoimmunotherapy and IO-chemo

Author response: We thank the reviewer for pointing this out. We have revised the manuscript to keep our terminology and labeling consistent.

2. Figure S1 shows that those with other types of NSCLC were excluded, but in table 1 there are 64 patients with other types of NSCLC. Did the authors detect any additional clinico-genomic predictors in this group?

Author response: Thank you for highlighting this discrepancy. The 64 cases with other types of NSCLC include adenosquamous (n=10), NSCLC not otherwise specified (n=48), large cell carcinoma (n=4), and sarcomatoid carcinoma (n=2). To keep our patient population consistent with our defined inclusion/exclusion criteria, we have removed the 6 patients with large cell + sarcomatoid histology, changing our total study population from 1139 to 1133.

With respect to the original 64 patients with 'other' histologies, the demographic information for this cohort is listed below. Genomically, this subgroup is enriched for *TP53* and potentially de-enriched for *STK11*. Unfortunately, analysis of clinico-genomic predictors were not powered in this small sub-cohort, but, as with squamous histology, we agree that more in-depth analysis in this patient population would be of interest. Consistent with the overall analysis, no significant differences were found in PFS and OS between ICI-mono and ICI-chemo in this cohort.

Parameters	Total No. (%)
Age at ICI started	
18-65	29 (45.3)
> 65	35 (54.7)
Gender	
Male	34 (53.1)
Female	30 (46.9)
Tobacco use	
Never	8 (12.5)
Former/current	56 (87.5)
Metastatic status at ICI start	
IVA	18 (28.1)
IVB	46 (71.9)
Liver metastasis at ICI start	
No	55 (85.9)
Yes	9 (14.1)
Brain metastasis at ICI start	
No	44 (68.8)
Yes	20 (31.2)
ICI Treatment	
ICI-mono	49 (76.6)
ICI-Chemo	15 (23.4)
Line of ICI therapy	
1 st	32 (50)
> 1 st	32 (50)
PD-L1 expression	
Low	9 (14.1)
Intermediate	11 (17.2)
high	16 (25)
Unknown	28 (43.8)

Reviewer Figure 9. Table depicting clinical features of patients with 'other' histology. A) Association of genomic alterations with 'other' histology. B) Progression-free (left) and overall survival (right) in patients with 'other' histology, stratified by treatment with ICI-mono vs ICI-chemotherapy.

3. Although tumour mutational burden is missing from this analysis which has been shown to correlate with response to immune checkpoint blockade, this is acknowledged by the authors in the discussion, and their use of smoking status as a clinical proxy. This could potentially be explored in further studies.

Author's response: We agree that this is an important limitation and one that we hope will be addressed in future analyses due to widespread implementation of much broader sequencing panels.

Reviewer #5 (Remarks to the Author): expert in lung cancer genomics

The authors examined real-world lung cancer patient data (n=1,139) who received ICI-chemo or ICI-mono therapy in routine cancer treatment to challenge the following questions (as described by the authors in

the introduction), which to date have not been adequately answered.

1. Who should receive combination therapy vs ICI monotherapy (ICI-mono)?
2. Clinicogenomic predictors of ICI response
3. Whether the addition of chemotherapy affects short- and long-term outcomes to ICI therapy in NSCLC?

For the issue 2, several gene alterations were picked up in the first-line cases (n=680): STK11, JAK2, and CDK6 with early progression with ICI-mono, and CDKN2A and CCND1 with worse long-term outcome with ICI-chemo.

In Task 3, ICI-mono was found to have a higher rate of progression at 3 and 6 months compared to ICI-chemo, but the long-term progression-free survival and overall survival were comparable. Based on these findings, the authors discuss that biomarkers such as STK11, JAK2, and CDK6 might be useful for identifying patients who may benefit from ICI-chemotherapy.

These findings are very interesting and may lead to a solution to Issue 1. This is a very interesting story based on real world data. However, STK11 seems to be associated with 3-month progression in both ICI-chemo and ICI-mono (Fig. 3A, B). Thus, the genomic predictors of Issue 1 are still not firmly determined.

They also found that the long-term prognosis is comparable between sequential and concurrent ICI and chemotherapy. This is interesting data, but needs to be validated in several cohorts before conclusions can be drawn.

In summary, the authors present some very interesting issues based on real-world data. However, they are not robust enough and do not have a strong enough impact to change routine oncology.

Author response: We thank the reviewer for this thoughtful summary and critique. There are two important specific points raised along with a more general concern regarding the robustness and impact of our findings. We address the specific points first:

1. Certain genomic predictors are significant in both the ICI-chemo and ICI-mono contexts. Thus, genomic predictors to adjudicate decision-making between ICI-chemo vs ICI-mono are not firmly established by our work.

One of the important implications of our analyses is that the determinants of long-term benefit to ICIs are most likely not specific to whether patients are treated with ICI-mono vs ICI-chemo, but rather, whether a patient has an ICI-responsive tumor. In the absence of a synergistic relationship between ICIs and chemotherapy, to some extent this result is what would be expected, as we know that chemotherapy itself only rarely induces long-term disease control. Consistent with this interpretation, many of the features that

we know associate with long-term anti-tumor immunity, e.g. PD-L1, associate with long-term PFS in both the ICI-mono and ICI-chemo treatment contexts. It is therefore not surprising that we also see that many genomic features are similarly predictive irrespective of treatment strategy.

However, there are two important additional points to be made. The analyses we performed do not imply that there is no role for chemotherapy, but rather allows us to define more specifically what that role is, which appears to be a therapeutic ‘hedge’ against early progression in patients who turn out not to have ICI-responsive tumors. Identifying those patients at risk for early progression on ICI-monotherapy may help ensure that they are given an effective therapy upfront, thereby mitigating the risk of rapid clinical deterioration that precludes treatment with chemotherapy in the second-line. If one takes the reviewer’s specific example of *STK11*, *STK11* does not emerge as differentially associated with benefit to ICI-chemo vs ICI-mono because the direction and magnitude of the effect of *STK11* is similar in both treatment contexts – regardless of treatment strategy, patients with *STK11* alterations are less likely to have long-term disease control. However, there may still be some benefit from treating these patients with combination therapy. If one analyzes the cumulative hazard of ICI-chemo vs ICI-mono over time in *STK11*-mutated patients, one can see that there is a significant benefit to ICI-chemo in the short-term:

Reviewer Figure 10. Additive hazard for PFS in ICI-chemotherapy vs ICI-monotherapy in patients with *STK11* alterations. Coefficient < 0 indicates improved PFS in ICI-chemo relative to ICI-mono.

Consistent with this overall approach, our integrative modeling for 3-month progression is much less predictive in the ICI-chemo context, likely because the additive effect of two treatments with non-overlapping resistance mechanisms render specific features less predictive. Thus, even if predictors such as *STK11* associate with worse outcomes to both ICI-mono and ICI-chemo – and, as emphasized above,

in the absence of treatment synergy this may to some extent be expected – a more careful definition of the role of chemotherapy may also help identify those patients who benefit from an upfront combination approach.

We do also want to highlight that the above discussion does not preclude there being more specific instances where the combination of ICI and chemotherapy does have unique biological effects, even if this does not seem to be the case in general. In our analyses, we identified *CDKN2A* alterations as having differential association with outcome to ICI-mono vs ICI-chemo in our primary cohort and a validation dataset. Other examples of relationships that have been proposed elsewhere include the potentiation of anti-tumor immune signaling when chemotherapy is given in the context of particular defects in DNA damage repair. To date, most of these relationships are more hypothetical than proven, and much work needs to be done to identify and subsequently validate that these treatment combinations truly change the biology of ICI response and resistance. Teasing out these very specific and nuanced relationships will require very large and well annotated cohorts, and we believe that studies such as ours contribute to this effort and are important in identifying potentially important relationships for further experimental and clinical validation.

2. The conclusion that long-term outcomes in ICI-chemo vs ICI-mono are comparable needs to be validated in several cohorts.

Thank you for highlighting this finding and the need for additional validation cohorts. We have obtained an independent internal validation cohort along with two external validation cohorts to address this question. Results from these three cohorts appeared comparable to our initial findings, with no difference in overall survival across all three groups nor in particular between patients treated with ICI-chemo vs those treated with ICI-mono followed by chemo. In all instances, there is a subset of ICI-mono only patients who have the shortest OS, consistent with a group of patients who progress on ICI-mono and are unable to be treated with subsequent chemotherapy. These validation analyses are included as **Supplementary Figure 12**, and are copied below as well:

Reviewer Figure 11. Analyses of overall survival (OS) stratified by 2nd-line therapy in the validation cohorts. A) MDACC validation cohort; B) Mayo cohort; and C) MGH cohort.

Finally, the reviewer argues that our findings are not robust enough to have a strong impact on routine oncologic practice. We believe we have significantly strengthened our findings in this revised manuscript by clarifying our overall statistical framework and validating our findings in multiple independent cohorts. We highlight in particular that our primary outcome analysis and the performance of the predictive model are all recapitulated in these validation cohorts, and we believe these analyses have considerably improved the robustness of our study.

More fundamentally, we fully agree that the questions we outline in this study require further study and validation, nor do we think that a single retrospective study – no matter how robust – can or should change clinical practice. However, we note that clinicians are already making these decisions with even less data to guide them, often relying on gestalt principals (higher disease burden, low PD-L1, etc) or incompletely proven concepts such as synergistic benefit. In this context, studies such as this can provide actual data which, when validated through further studies or clinical trials, can meaningfully influence clinical practice. In the interim, we believe that this study helps clarify a number of important conceptual issues, highlighting that chemotherapy enhances short rather than long-term outcomes without evidence of synergy, and we are the first manuscript to propose and validate a clear set of clinical and basic genomic features that might contribute to a higher likelihood of early progression, therefore providing a helpful framework to guide the decision toward chemo-ICI. We therefore expect that these results can provide actual data that informs the decisions that are already being made, and more concretely, we anticipate that this work can serve as the basis for more formalized and validated nomograms or other decision support tools that can be directly integrated into clinical practice.

References

1. Pérol, M. *et al.* Effectiveness of PD-(L)1 Inhibitors Alone or in Combination With Platinum-Doublet Chemotherapy in First-Line (1L) Non-Squamous Non-Small Cell Lung Cancer (nsq-NSCLC) With PD-L1-High Expression Using Real-World Data. *Ann. Oncol.* **0**, (2022).
2. Aalen, O. O. A linear regression model for the analysis of life times. *Stat. Med.* **8**, 907–925 (1989).
3. Ballarini, N. M., Thomas, M., Rosenkranz, G. K. & Bornkamp, B. subtee: An R Package for Subgroup Treatment Effect Estimation in Clinical Trials. *J. Stat. Softw.* **99**, 1–17 (2021).
4. Miao, D. *et al.* Genomic correlates of response to immune checkpoint blockade in microsatellite-stable solid tumors. *Nat. Genet.* **50**, 1271–1281 (2018).
5. Rizvi, H. *et al.* Molecular Determinants of Response to Anti-Programmed Cell Death (PD)-1 and Anti-Programmed Death-Ligand 1 (PD-L1) Blockade in Patients With Non-Small-Cell Lung Cancer Profiled With Targeted Next-Generation Sequencing. *J. Clin. Oncol.* **36**, 633–641 (2018).
6. Vokes, N. I. *et al.* Harmonization of Tumor Mutational Burden Quantification and Association With Response to Immune Checkpoint Blockade in Non-Small-Cell Lung Cancer. *JCO Precis. Oncol.* 1–12 (2019) doi:10.1200/PO.19.00171.
7. Doroshov, D. B. *et al.* Programmed Death-Ligand 1 Tumor Proportion Score and Overall Survival From First-Line Pembrolizumab in Patients With Nonsquamous Versus Squamous NSCLC. *J. Thorac. Oncol.* S1556086421023923 (2021) doi:10.1016/j.jtho.2021.07.032.
8. Osta, B. E. *et al.* Characteristics and Outcomes of Patients With Metastatic KRAS-Mutant Lung Adenocarcinomas: The Lung Cancer Mutation Consortium Experience. *J. Thorac. Oncol.* **14**, 876–889 (2019).
9. Arbour, K. C. *et al.* Treatment Outcomes and Clinical Characteristics of Patients with KRAS-G12C-Mutant Non-Small Cell Lung Cancer. *Clin. Cancer Res.* **27**, 2209–2215 (2021).
10. Liu, C. *et al.* The superior efficacy of anti-PD-1/PD-L1 immunotherapy in KRAS-mutant non-small cell lung cancer that correlates with an inflammatory phenotype and increased immunogenicity. *Cancer Lett.* **470**, 95–105 (2020).
11. Skoulidis, F. *et al.* STK11/LKB1 Mutations and PD-1 Inhibitor Resistance in KRAS-Mutant Lung Adenocarcinoma. *Cancer Discov.* **8**, 822–835 (2018).
12. Mazieres, J. *et al.* Immune checkpoint inhibitors for patients with advanced lung cancer and oncogenic driver alterations: results from the IMMUNOTARGET registry. *Ann. Oncol.* **30**, 1321–1328 (2019).
13. Herbst, R. S. *et al.* LBA4 Association of KRAS mutational status with response to pembrolizumab monotherapy given as first-line therapy for PD-L1-positive advanced non-squamous NSCLC in Keynote-042. *Ann. Oncol.* **30**, xi63–xi64 (2019).
14. Huang, Q. *et al.* Impact of PD-L1 expression, driver mutations and clinical characteristics on survival after anti-PD-1/PD-L1 immunotherapy versus chemotherapy in non-small-cell lung cancer: A meta-analysis of randomized trials. *Oncol Immunology* **7**, e1396403 (2018).
15. Schmidt, E. V. *et al.* Assessment of Clinical Activity of PD-1 Checkpoint Inhibitor Combination Therapies Reported in Clinical Trials. *JAMA Netw. Open* **3**, e1920833 (2020).
16. Palmer, A. C. & Sorger, P. K. Combination cancer therapy can confer benefit via patient-to-patient variability without drug additivity or synergy. *Cell* **171**, 1678–1691.e13 (2017).

REVIEWERS' COMMENTS

Reviewer #2 (Remarks to the Author):

I have no further comments; the authors incorporated all my comments/suggestions.

Reviewer #3 (Remarks to the Author):

The authors have substantially expanded studies from the original manuscript that include the addition of new datasets, analyses, methods etc. I am satisfied with this revision and the rebuttal to my original concerns.

Reviewer #4 (Remarks to the Author):

The authors have provided a comprehensive and data-driven response to the reviewer comments.

Reviewer #5 (Remarks to the Author):

The manuscript has been improved by adding data of additional cohorts and discussion. I understand that this is a rather challenging topic regarding the treatment of ICI. This manuscript will be an important resource for further research in the near future.

REVIEWERS' COMMENTS

Reviewer #2 (Remarks to the Author):

I have no further comments; the authors incorporated all my comments/suggestions.

Reviewer #3 (Remarks to the Author):

The authors have substantially expanded studies from the original manuscript that include the addition of new datasets, analyses, methods etc. I am satisfied with this revision and the rebuttal to my original concerns.

Reviewer #4 (Remarks to the Author):

The authors have provided a comprehensive and data-driven response to the reviewer comments.

Reviewer #5 (Remarks to the Author):

The manuscript has been improved by adding data of additional cohorts and discussion. I understand that this is a rather challenging topic regarding the treatment of ICI. This manuscript will be an important resource for further research in the near future.

Author's response to all reviewers: We would like to thank you for taking the necessary time and effort to review the manuscript. We sincerely appreciate all your valuable comments and suggestions, which helped us in improving the quality of the manuscript.